# Fungal antigenic variation using mosaicism and reassortment of subtelomeric genes' repertoires

Caroline S. Meier [1], Marco Pagni [2], Sophie Richard[1], Konrad Mühlethaler[3], João M. G. C. F. Almeida [4], Gilles Nevez [5,6], Melanie T. Cushion [7,8], Enrique J. Calderón[9,10] & Philippe M. Hauser [1] ✉

Surface antigenic variation is crucial for major pathogens that infect humans. To escape the immune system, they exploit various mechanisms. Understanding these mechanisms is important to better prevent and fight the deadly diseases caused. Those used by the fungus *Pneumocystis jirovecii* that causes life-threatening pneumonia in immunocompromised individuals remain poorly understood. Here, though this fungus is currently not cultivable, our detailed analysis of the subtelomeric sequence motifs and genes encoding surface proteins suggests that the system involves the reassortment of the repertoire of ca. 80 non-expressed genes present in each strain, from which single genes are retrieved for mutually exclusive expression. Dispersion of the new repertoires, supposedly by healthy carrier individuals, appears very efficient because identical alleles are observed in patients from different countries. Our observations reveal a unique strategy of antigenic variation. They also highlight the possible role in genome rearrangements of small imperfect mirror sequences forming DNA triplexes.

The fungus *Pneumocystis jirovecii* is an obligate biotrophic parasite that colonizes specifically the human lungs[1]. In immunocompromised patients, mostly HIV positive patients and transplant recipients, it causes a life-threatening pneumonia that is among the most frequent invasive fungal infections[2]. Currently, the study of *P. jirovecii* biology is difficult due to the lack of a long-term in vitro culturing method[3].

This fungus lacks chitin, α-glucans, and outer chain N-mannans commonly present in fungal walls, which may help escape the host's immune responses during colonization[4]. In addition, like other major

microbes pathogenic to humans, it possesses a system of surface antigenic variation that appears essential for survival because it represents up to 6% of its highly compacted genome[5–7]. The most important player of this system is a superfamily including six families of major surface glycoproteins (Msg-I to VI[4,8]). These are supposed to be responsible for adherence to various human proteins present in the lungs and on macrophages[9–12]. All Msgs are believed to cover asci and trophic cells that are present during proliferation, except those of family VI which could be present only at the surface of ascospores, as

[1]Institute of Microbiology, Lausanne University Hospital and University of Lausanne, Lausanne, Switzerland. [2]Vital-IT Group, SIB Swiss Institute of Bioinformatics, Lausanne, Switzerland. [3]Institute for Infectious Diseases, University of Bern, Bern, Switzerland. [4]UCIBIO, Applied Molecular Biosciences Unit, Department of Life Sciences, NOVA School of Science and Technology, Universidade NOVA de Lisboa, 2829–516 Caparica, Portugal. [5]Laboratoire de Parasitologie et Mycologie, Hôpital de La Cavale Blanche, CHU de Brest, Brest, France. [6]Infections respiratoires fongiques (IFR), Université d'Angers, Université de Brest, Brest, France. [7]Department of Internal Medicine, Division of Infectious Diseases, College of Medicine, University of Cincinnati, Cincinnati, OH 45267, USA. [8]Cincinnati VAMC, Medical Research Service, Cincinnati, OH 45220, USA. [9]Instituto de Biomedicina de Sevilla, Hospital Universitario Virgen del Rocío/Consejo Superior de Investigaciones Científicas/Universidad de Sevilla, Seville, Spain. [10]Centro de Investigación Biomédica en Red de Epidemiología y Salud Pública, Servicio de Medicina Interna, Hospital Universitario Virgen del Rocío, Departamento de Medicina, Facultad de Medicina, Seville, Spain. ✉e-mail: Philippe.Hauser@chuv.ch

is the case in *Pneumocystis murina* infecting specifically mice[13]. Family I is the most abundant in number of genes, transcripts, and proteins at the cell surface[4,14,15]. During pneumonia, families I and III represent respectively ca. 85 and 10% of the *msg* transcripts, whereas the other families only 1% each[15]. The genes encoding Msgs are located within all subtelomeres of the 17–20 chromosomes of *P. jirovecii*, the genes of family I being closest to the telomeres[4,8]. The subtelomeric localization favors ectopic recombinations within the meiotic bouquet of telomeres, gene silencing, and possibly mutagenesis[16]. In addition, recombinations of genes in this genomic region is advantageous as it has no, or little, on the overall chromosome structure[6]. Genes of families II–VI each have their own promoter and could be constitutively and simultaneously expressed[15]. We observed that several alleles of each family, except V and VI, are expressed during a single infection. However, it is not known if all or only some genes of each of these families are expressed in single cells. This regulation might occur during the cell cycle or by silencing because of the proximity of the telomeres (the "telomere position effect")[16]. On the other hand, one gene out of the ca. 80 genes of family I is believed to be expressed at a time in a cell thanks to the presence of a single copy promoter in the genome, within the so called upstream conserved sequence (UCS)[17,18]. At the end of the UCS, a 33 bps-long sequence is present, the conserved recombination junction element (CRJE), which is also found at the start of each of all *msg*-I alleles[19]. It is most probably the preferred site of recombination allowing the exchange of the downstream expressed allele. The spontaneous exchange of the single *msg*-I gene expressed per cell is thought to create subpopulations of cells, each expressing a different *msg*-I allele. A second potential mechanism of antigenic variation relies on intragenic recombinations of the *msg* gene sequences, i.e. gene mosaicism[4,8,18,20].

The present work aimed at better understanding the mechanisms involved in the antigenic variation system of *P. jirovecii* by investigating the repertoires of the *msg*-I genes present in patients from different geographical locations. Our observations allow us to propose a model for the surface antigenic variation system of *P. jirovecii*.

## Results

### Amplification and identification of the *msg*-I alleles

We amplified the *msg*-I alleles present in the BALs of 24 patients with *Pneumocystis* pneumonia from five different geographical locations (Supplementary Table 1). Generic primers were used in two different PCRs to amplify specifically either all entire *msg*-I genes both expressed and non-expressed (hereafter called the "genomic repertoire"), or all entire expressed *msg*-I genes (the "expressed repertoire"). The PCR products were sequenced using PacBio circular consensus sequence (CCS). The reads were processed using a specifically developed bioinformatics pipeline dedicated to the identification of the different biological alleles present, as well as to the determination of their abundance. The known issues represented by chimeric sequences created during PCR amplification and PacBio sequencing errors were specifically addressed and are not believed to affect the results presented below (see "Methods"). The diversity of the alleles reported might however be underestimated.

### *msg*-I alleles identified in the patients

Among the 48 PCR products from the 24 patients, 1007 distinct *msg*-I alleles were identified. They had a mean pairwise sequence identity of 65.7% ± SD 9%. This value is in agreement with that based on 11 alleles from Switzerland that we published previously (71% ± SD 7%)[8], as well as with that of 70.5% ± SD 3% among a collection of 80 *msg*-I genes from USA[4]. If we define pseudogenes as alleles with at least one stop codon, they represented 5.6% of the 1007 alleles. This is consistent with the single previous observation of 11% among 28 genes with a CRJE[8]. Thus, 94.4% of the 1007 alleles presented a fully open reading frame without any introns.

The 24 genomic and 24 expressed repertoires included respectively 917 and 538 distinct alleles that were sorted using hierarchical classification trees (Fig. 1). Both trees highlighted the presence of two major and one smaller subgroups of the *msg*-I alleles, which is consistent with previous observations[4,8,20]. The two major subgroups constitute a single family, based on the occurrence of recombinations between them[8,20]. The smaller one corresponds to outlier *msg* genes that could not be yet classified into family I[8]. The significance of these subgroups remains unexplained so far.

### Repertoires of *msg*-I alleles present in the patients

The alleles of both the expressed and genomic repertoires present in each patient were uniformly spread along the trees of the alleles (Fig. 1). There were no clades of *msg*-I sequences associated with specific repertoires, suggesting that sample provenance did not explain the tree topology. The genomic repertoires contained 44–185 alleles per patient (104 ± SD 40), whereas 2–108 alleles were present in the expressed repertoires (37 ± SD 34, Supplementary Table 5). The number of alleles of the genomic and expressed repertoires were not significantly associated with the fungal loads present in the samples (Pearson correlation respectively 0.15 and 0.39, *p*-value = 0.50 and 0.08, *n* = 22; data in Supplementary Tables 1 and 5). Notably, three out of the five samples from Brest (BR1, BR2, BR3) harbored the least diverse expressed repertoires with 2, 3, and 5 alleles, but no correlation with the fungal load or underlying disease was observed.

The variation of the number of alleles of the genomic repertoires was at least partially explained by the number of *P. jirovecii* strains present in the patients. Indeed, a significant correlation was observed between these numbers (Fig. 2). On the other hand, the expressed repertoires showed no correlation, suggesting the involvement of other more important parameters. A reproducibility experiment showed that the varying abundances of the alleles of the genomic repertoire harbored by patient LA2 infected by a single strain were reproducible in duplicate analyses (Supplementary note 1, Supplementary Table 6, Supplementary Fig. 4). This was unexpected because these alleles are all present at a single copy per genome and thus in equal abundance in the DNA sample. This might result from (i) a varying efficiency of amplification and/or PacBio sequencing of the alleles, some being not amplifiable at all, and/or (ii) an undetected co-infecting strain present in low abundance. The first hypothesis is more likely because an underestimation of the numbers of alleles would explain better the results. Indeed, it would explain that the patients assessed to be infected by a single strain using a second genotyping method harbored in their genomic repertoire less than the 80 alleles reported in the literature (44, 49, 56 in respectively CI5, LA2, BE2; Supplementary Note 2, Supplementary Table 7). It would also explain that four of the other six patients that turned out to harbor two strains rather than one harbor only 54, 59, 61, and 79 alleles (respectively LA7, LA9, SE1, and LA3). Importantly, this underestimation would not impact the conclusions drawn because none is based on the absolute values of these numbers.

Due to the design of the two PCRs, the expressed repertoire should be a subset of the genomic repertoire for each patient. Consistently, high proportions of the alleles of the expressed repertoires were also present in the corresponding genomic repertoires (85% ± SD 17%, 40–100%, "% expressed in genomic", Supplementary Table 5). Eighteen out of the 24 patients presented a proportion lower than 100% probably because of the limitation of the methodology (see above). Conversely and as expected, lower proportions of the alleles of the genomic repertoires were present within the corresponding expressed repertoires (33% ± SD 32%, 2 to 100%, "% genomic in expressed", Supplementary Table 5). These latter proportions are consistent with the single previous estimation of 50% among 28 genes[8], they correspond to the alleles that are both non-expressed and expressed.

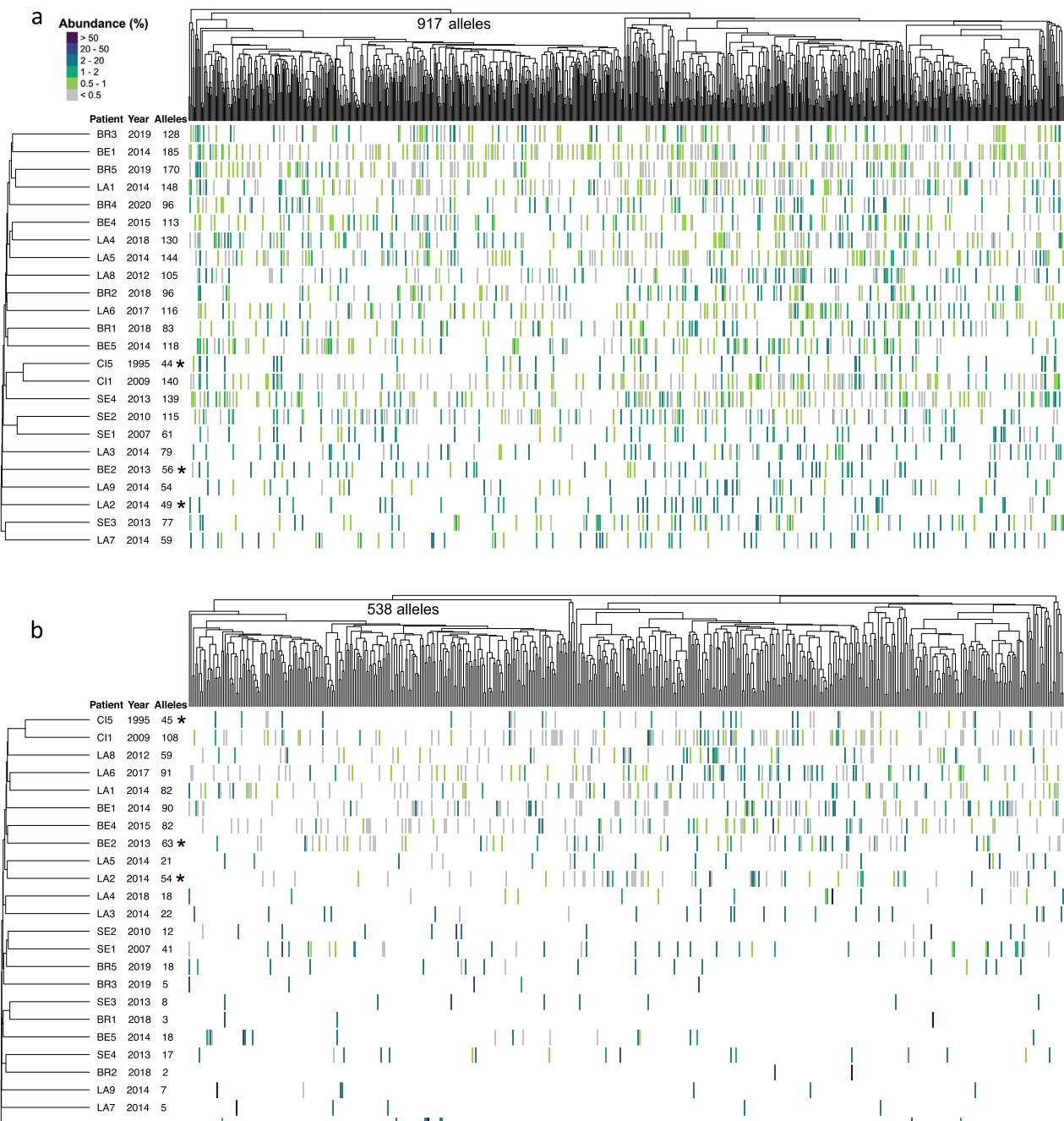

**Fig. 1 | Composition of the genomic and expressed *msg*-I repertoires.** The genomic and expressed repertoires are shown respectively in panels (**a**) and (**b**). Each vertical line of the heatmap represents an allele present in the given repertoire with the color figuring its abundance in % of all reads composing the repertoire, as indicated at the top left of (**a**). The 917 and 538 distinct alleles identified in the repertoires of the 24 patients were sorted using hierarchical classification trees of the multiple alignment of the allele sequences (Fitch distance, average linkage). The black stars identify the three patients assessed to be infected by a single strain (Supplementary Note 2). The patients were sorted using a tree of presence/absence of each allele in their repertoire (binary distance, average linkage). The collection year and the number of alleles present in the patient are indicated next to the patient's name. LA Lausanne, BE Bern, BR Brest, CI Cincinnati, SE Seville. Data are provided in Supplementary Data 5.

### Similarity of the *msg*-I repertoires between the patients

The genomic and expressed *msg*-I repertoires present in the patients were also sorted using hierarchical classification trees (Fig. 1). The inspection of these trees reveals that each repertoire was notably different from all the others. This distinctiveness implies the absence of obvious correlation of the repertoires with the year or city of collection of the sample, the underlying disease affecting the patient, or the *P. jirovecii* genotype(s) causing the infection. The genomic repertoires of two samples from Seville, SE1 and SE2, as well as the genomic and expressed repertoires of the two samples from Cincinnati, CI1 and CI5, were possible exceptions as they were slightly related.

Although they were all distinct, the repertoires shared many alleles. Indeed, 84% ± SD 7% of the alleles of each genomic repertoire were present in at least one genomic repertoire of another patient. Of note, this level of sharing might be imprecise because of the trimming of the sequences and the possible underestimation of the number of

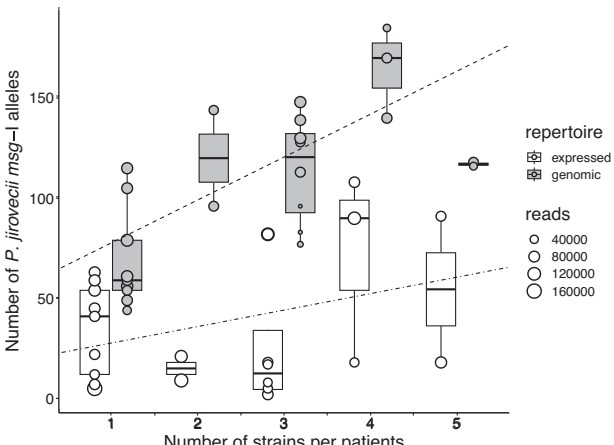

**Fig. 2 | Correlation between the number of *P. jirovecii* strains and the number of alleles.** For the genomic repertoires of the 24 patients, the correlation between the number of alleles and the number of stains was 0.74 (Pearson correlation weighted by the number of reads, *p*-value = 4.2 × 10−5, *n* = 24); the slope of the regression line and intercept were 21.48 and 56.11, respectively. For the expressed repertoires the correlation was 0.32 (*p*-value = 0.12); the regression slope and intercept were 19.40 and 8.24, respectively. The size of each point indicates the number of reads generated by PacBio CCS for each sample. No correlation was observed between the number of reads and the number of different alleles identified in each sample (Pearson correlation, *p*-value: 0.20, correlation −0.19, SD 0.14). The center lines of the boxplots represent the median values, the box limits the 25th and 75th percentiles, and the whiskers extend to the largest values no further than 1.5 × interquartile range. Source data are provided in the Source Data file.

alleles present in the genomic repertoires (see previous section). However, this would not impact the conclusions drawn as they are not based on the absolute values. The value was 61% ± SD 19% for the expressed repertoires. Figure 3 gives the proportion of alleles shared by each repertoire and allow visualizing that the overlaps of the repertoires did not differ significantly according to the city and continent, as well as of the year of collection.

### Distribution of the *msg*-I alleles among the patients

Approximately half of the 917 alleles present in the 24 genomic repertoires were found in only one city, of which the vast majority were in a single patient (88.0%, 411 among 467, Table 1 in Supplementary Data 6). Thus, each of the 24 patients harbored several of the 411 alleles that were observed in only one patient. The remaining half was present in more than one city, 3.6% occurring even in all five cities. The proportion of the alleles found in only one city among the 24 expressed repertoire was higher than in the genomic repertoires (72.9%), but a similarly high proportion of which was found in a single patient (88.5%, 347 among 392, Table 1 in Supplementary Data 6). A single expressed allele was present in all five cities (0.2%). Figure 4 allows the visualization of these proportions and shows that comparable results were observed in each of the five cities.

In order to understand the parameters influencing the distributions of the alleles among the patients, we performed two simulations experiments in silico. First, we investigated the size of the reservoir from which the alleles are retrieved. We simulated the 24 genomic repertoires by drawing 24 times the 104 alleles present on average in them out of a simulated reservoir comprising 1000–5000 alleles. We repeated this draw 30 times, and then determined the mean number of draws of each allele. The distribution obtained with the reservoir including 2000 alleles was most similar to that observed in the genomic repertoires of our data (Supplementary Fig. 5a). Similarly, drawing 24 times 37 alleles to simulate the 24 expressed repertoires, the distribution with a reservoir of 1000 alleles resembled that we observed for the expressed repertoires (Supplementary Fig. 5b). In the second

simulation experiment, we determined the effect of analyzing less than the 24 patients. We drew 30 times 5, 10, 15, or 20 patients randomly, and calculated the mean numbers of alleles observed. Consistent with the first simulation experiment, the numbers of alleles increased regularly, showing that a plateau corresponding to the complete reservoir was not reached with the analysis of 24 patients (Supplementary Fig. 6). These simulations suggest that the high proportion of alleles we observed only once in single patients could be explained by a large reservoir of alleles. However, a not mutually exclusive hypothesis is that a proportion of them corresponds to new alleles created by mosaicism within each patient because this is probably necessary for the survival of the fungus.

### Abundance of the *msg*-I alleles in the patients

Each population of *P. jirovecii* cells is expected to be composed of subpopulations, each expressing a distinct *msg*-I allele. The size of these subpopulations, and thus the abundance of the expressed alleles, may vary according to a possible advantage over the host immune system, or to other parameters. To test this hypothesis, the abundance of an allele was defined as its number of reads in percent of the total number of those present in the given repertoire. These abundances were confirmed using subcloning in the wet-lab (Supplementary Note 3, Supplementary Table 8). All the 24 genomic repertoires showed a underdispersed distribution of the allele abundances which remains under a maximum of 8% (Fig. 5). By contrast, 22 out of the 24 expressed repertoires, i.e. except those of LA6 and BE1, showed a overdispersed distribution of these abundances, with up to 6 highly abundant alleles (≥8%), and 107 low abundant alleles (<8%). The highest abundance was seen in patient LA7 at 71.9%. Among the 62 expressed alleles with an abundance ≥8%, 40 were observed in more than one patient (64.5%). This contrasted drastically with the same proportion among all expressed alleles, including those at <8% (35.5%, 100 − 64.5%, Supplementary Data 5). This difference suggests that the alleles at ≥8% might have presented a selective advantage, for example over the host immune system of the individuals that harbored them.

The technical variability of the expressed repertoires characterisation is the major limitation of the present study (Supplementary Note 1, Supplementary Table 6, Supplementary Fig. 4). Nevertheless, the overdispersion of the allele abundances concerned both duplicates of all four expressed repertoires that we analyzed (Supplementary Fig. 4b). Moreover, all 11 alleles with an abundance ≥8% in at least one duplicate were present in both duplicates, six of them being at <8% in the other duplicate. Thus, despite the variation of the expressed abundances due to the limitation of the method, the distributions of the abundances differed clearly between the genomic and expressed repertoires. This further supports the mutually exclusive expression hypothesis. Indeed, the expressed alleles are likely to correspond to subpopulations expressing each a specific allele, the most frequent alleles by the largest ones.

### Sequences flanking the *msg* genes

The *msg*-I genes present short conserved sequences of a length of ca. 30 bps before and after their CDS (coding sequence), the CRJE and 31 bps located after the stop codon, in which we placed our primers for amplification of the genomic repertoires. To investigate if similar conserved sequences flank the *msg* CDSs of the other families, we took advantage of the 37 subtelomeres that we previously assembled from a single strain using the PacBio sequencing technology (Supplementary Figs. 7 and S8)[8]. These subtelomeres carry 113 distinct *msg* genes of the six *msg* families. We used the 200 bps located immediately up- or downstream of 20 representative CDSs as queries in BLASTn analyses against the PacBio *P. jirovecii* genome assembly. Most of the genes of families I–V produced numerous significant hits with other sequences that flanked a CDS of a gene or pseudogene of

the same family, most often both up- and downstream (Table 2 in Supplementary Data 6, Supplementary Data 1, 2, 3, 4 and 5). The hits represented variable proportions of the up- or downstream sequences of the same family present in the assembly (20–100%).

Importantly, for families I and IV, the identities of these hits with the query were ca. 10% higher than those between the genes themselves (Supplementary Data 2). On the other hand, these identities were similar between families II, III, and V. Notably, for family I, fewer hits

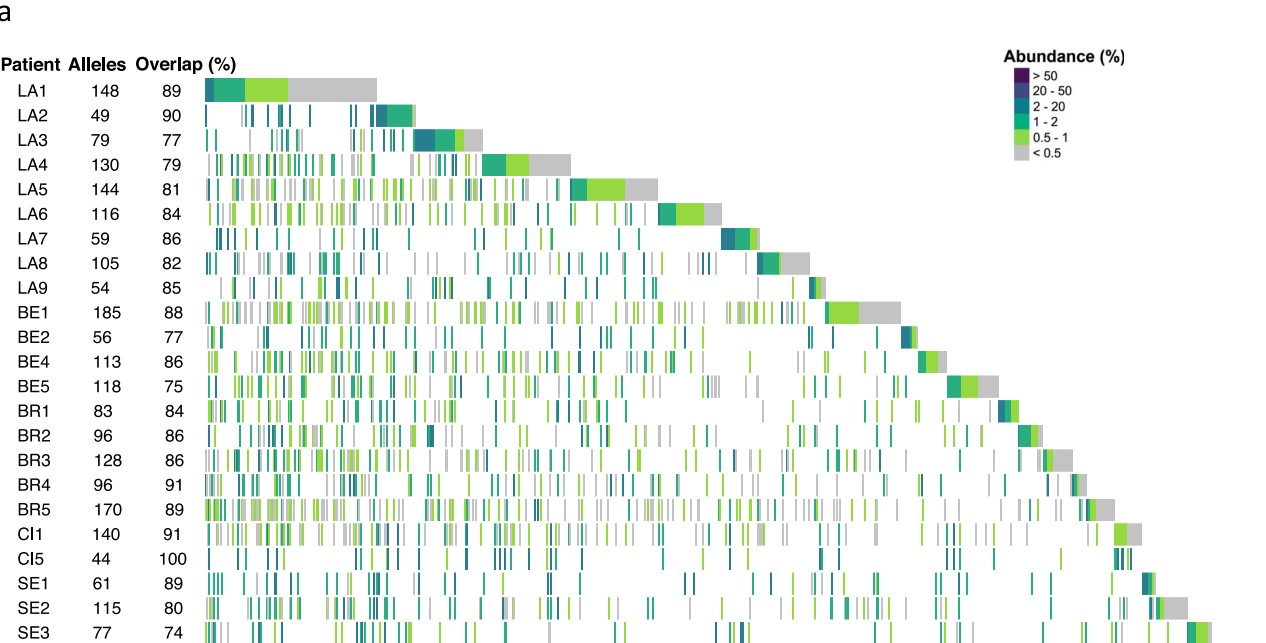

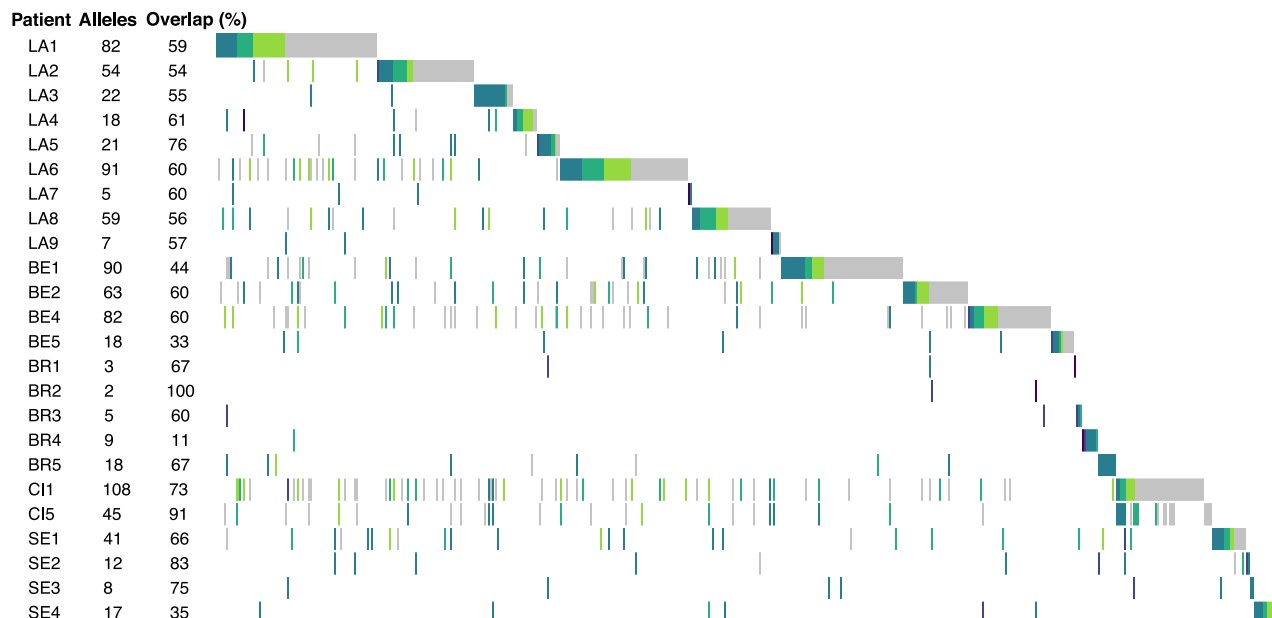

**Fig. 3 | Genomic and expressed repertoires of the 24 patients ordered by city.** The genomic and expressed repertoires are shown respectively in panels (**a**) and (**b**). This order allows visualize the alleles that the patients share. The order of the cities was arbitrarily chosen. Each vertical line represents an allele present in the given repertoire with the color figuring its abundance in % of all reads composing the repertoire, as indicated at the top right of (**a**). For each patient, the alleles that are stacked on the right correspond to those not shared with the patient(s) that are placed above. The number of alleles present in each repertoire and the proportion of alleles shared with other patient(s), i.e. the overlaps, are indicated next to the patient's name. LA Lausanne, BE Bern, BR Brest, CI Cincinnati, SE Seville. Data are provided in Supplementary Data 5.

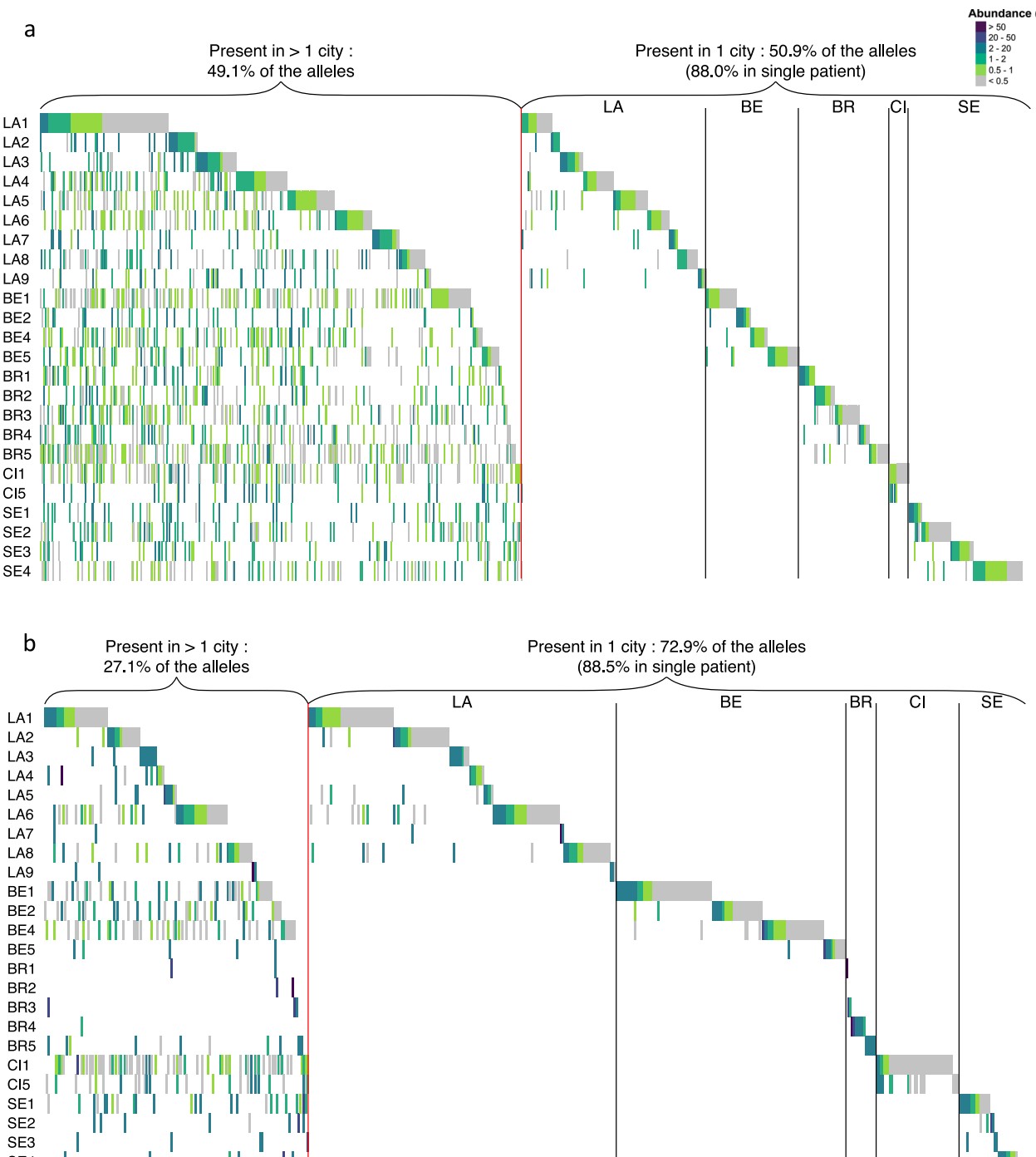

**Fig. 4 | Genomic and expressed repertoires of the 24 patients ordered by city and presence of the alleles in one or multiple cities.** The genomic and expressed repertoires are shown respectively in panels (**a**) and (**b**). The order of the cities was arbitrarily chosen. Each vertical line represents an allele present in the given repertoire with the color figuring its abundance in % of all reads composing the repertoire, as indicated at the top right of (**a**). LA Lausanne, BE Bern, BR Brest, CI Cincinnati, SE Seville. Data are provided in Supplementary Data 5.

were found for the pseudogenes than for the genes (7–38% versus 67–79 %). Inspection of the alignments with the hits for families II–V did not identify any conserved sequences flanking the CDSs, contrary to family I. These analyses revealed that the up- and downstream regions of *msg* CDSs are often similar among the genes of the same family, except for family VI, and that those of families I and IV are even more conserved than the genes nearby.

These analyses also revealed that the upstream sequences of the genes of families II and III are often similar, but not their downstream sequences (Supplementary Data 6). The identities between these hits

and their query were much higher than those between the genes nearby (Supplementary Data 2, mean difference of 29.8% ± SD 5.9%). These hits included an important proportion of the upstream sequences of the other family (50 to 78%). Besides, genes no. 3, 37, 55 and 8 of these two families showed no or lower similarities with the other family (Supplementary Table 2). The latter genes are located centrally in the subtelomeres, whereas the others are located at their extremities (genes no. 7, 25, 34, 53, Supplementary Fig. 7). Further-more, the similarity between the 200 bps upstream of genes of families II and III proved to extend within the CDS on ca. 100 bps

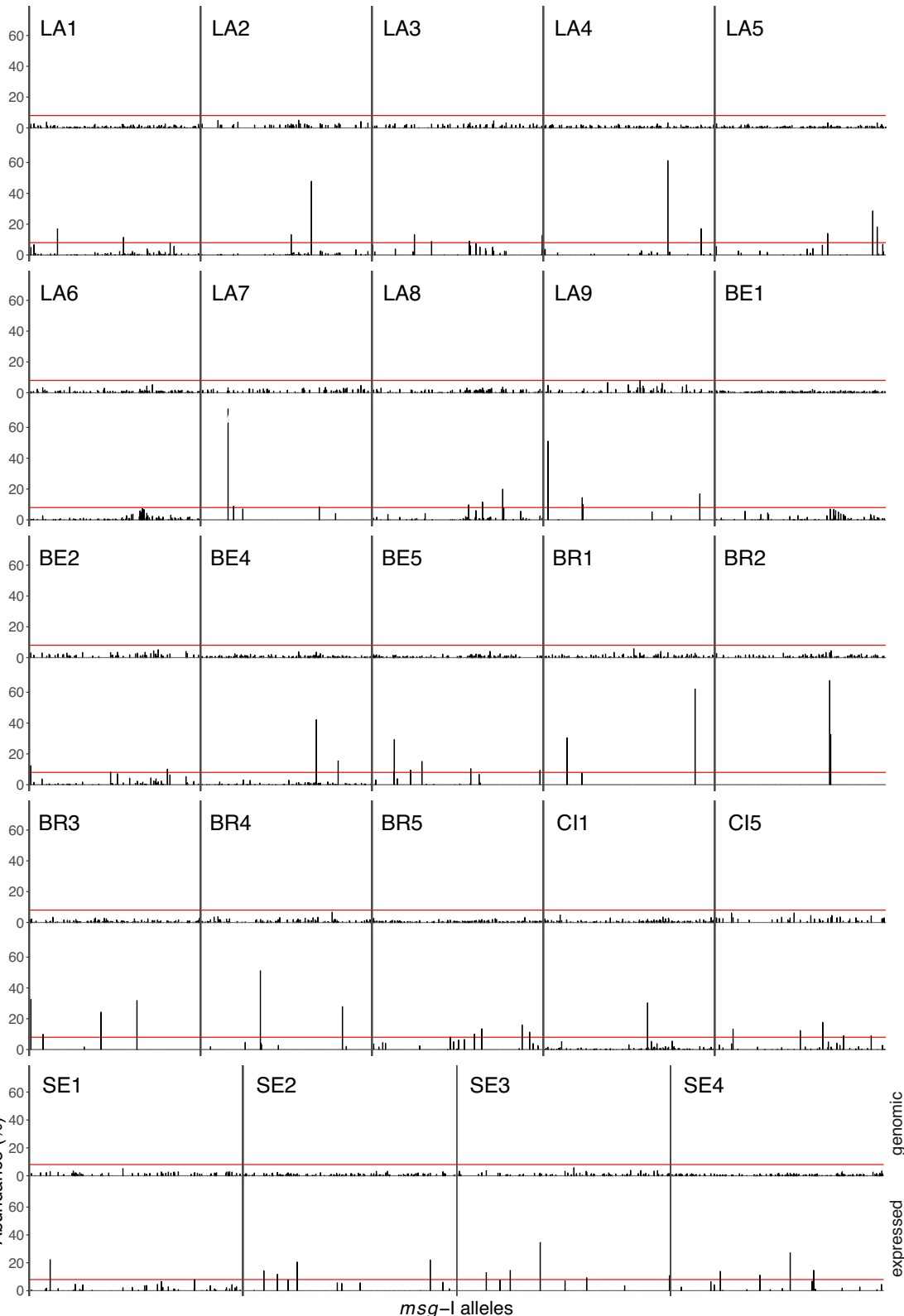

**Fig. 5 | Abundance of the alleles present in the genomic (top row) and expressed (bottom row) repertoires.** The alleles were sorted using a hierarchical classification tree of a multiple alignment of all 1007 allele sequences found in the 24 patients (Source data are provided in the Source Data file). The red lines indicate an abundance of 8.0%. LA Lausanne, BE Bern, BR Brest, CI Cincinnati, SE Seville.

(Supplementary Fig. 9). In conclusion, the upstream regions of the CDSs of *msg* families II and III are often similar and more conserved than the genes nearby, in particular among genes located at the distal tip of the subtelomeres, but not their downstream regions.

## Mosaicism of the *msg* genes

The mosaicism of *msg* genes, i.e. being composed of fragments potentially originating from other genes, was suggested based on the detection of recombinations and duplicated fragments strictly among

members of the same family[8,20]. We investigated the possible mosaicism of the 917 *msg*-I alleles identified in the genomic repertoires during the present study. We used the BLASTn algorithm because it allows the analysis of a large set of alleles. Inspection of the alignments of three randomly chosen alleles with their closest hit in BLASTn comparison identified in each case four to eight duplicated fragments of a size ≥100 bps (total of 16 fragments, one alignment is shown in Supplementary Fig. 10). We then searched each fragment within the 917 alleles using again BLASTn comparison. It appeared that the fragments were conserved among closely related alleles, i.e. within the same subgroup of alleles of the tree (Fig. 6). They were conserved in several patients, regardless of the year of the pneumonia episode, or city and continent of origin. Moreover, the two fragments <200 bps were present within two different subgroups of alleles (blue and purple in Fig. 6). In the single case of the fragment labeled in purple, the two subgroups corresponded to very distant alleles because they belonged to the two major subgroups of *msg*-I alleles (see above, section "*msg*-I alleles identified in the patients"). By contrast, the fragment ≥200 bps shown in red distributed in only one subgroup. Identical distributions in function of the size of the duplicated fragments were observed for all 16 duplicated fragments, ten ≥200 bps and six <200 bps, two among the latter being nevertheless present only in a single subgroup of alleles.

To understand the phenomenon better, we searched thoroughly duplicated fragments within the 10 representative subtelomeres of the 37 previously assembled[8] shown in Supplementary Fig. 7. The results confirm the mosaicism of the *msg* genes and pseudogenes previously reported[8], reveal that the intergenic spaces are also concerned, and suggest that no hotspots of recombination exist along the *msg* genes (Supplementary Note 4, Supplementary Table 9, Supplementary Figs. 7, 8 and 11).

## Structure of the CRJE sequence present at the beginning of each *msg*-I gene

The 33 bps CRJE sequence that is conserved at the beginning of each *msg*-I gene is probably the site of recombination allowing the exchange of the expressed allele. It is specific to *P. jirovecii*, and we did not identify any site-specific recombinase that could target it (Supplementary Notes 5 and 6). We identified two important features of the CRJE.

First, each strand of the CRJE is enriched in purines or pyrimidines that are part of an imperfect mirror repeat (Fig. 7a). An AT bp is present between the copies of the mirror repeats (position 14), and four AT bps are organized as inverted repeats at the end of the repeat, at positions 26–29 (TTAA). According to a body of literature[21–24], the features of the CRJE suggest that it can form two isomers of a non-canonical H-DNA, so called *H-DNA (Fig. 7b). *H-DNA is an intramolecular DNA triplex that, in this case, would be made of 11 base triads involving Hoogsteen bonds and presenting a single-stranded stretch of 12 bps (Supplementary Fig. 12a, b shows the structure of the base triads and a 3D model of DNA triplex). Although slightly smaller, the CRJE sequence closely resembles the canonical sequences that have been reported to form *H-DNA (Supplementary Fig. 12c). Consistently, seven out of the 11 base triads potentially formed by the CRJE are one of the two most frequent reported to constitute *H-DNA (CG*G, the symbol * in the triad stands for Hoogsteen bonds). Two of the 11 triads have been reported only in H-DNA so far (CG*C, GC*G), whereas the two remaining GC*C are non-canonical requiring more energy to be integrated in a DNA triplex but which have been observed in vitro. *H-DNA presents sequence requirements much less stringent than H-DNA[21], the mirror repeat may even not be present, so that the specificities of the CRJE sequence and the alternate triads that it may form are plausible. DNA triplexes are believed to play a number of roles in the cell[22,23].

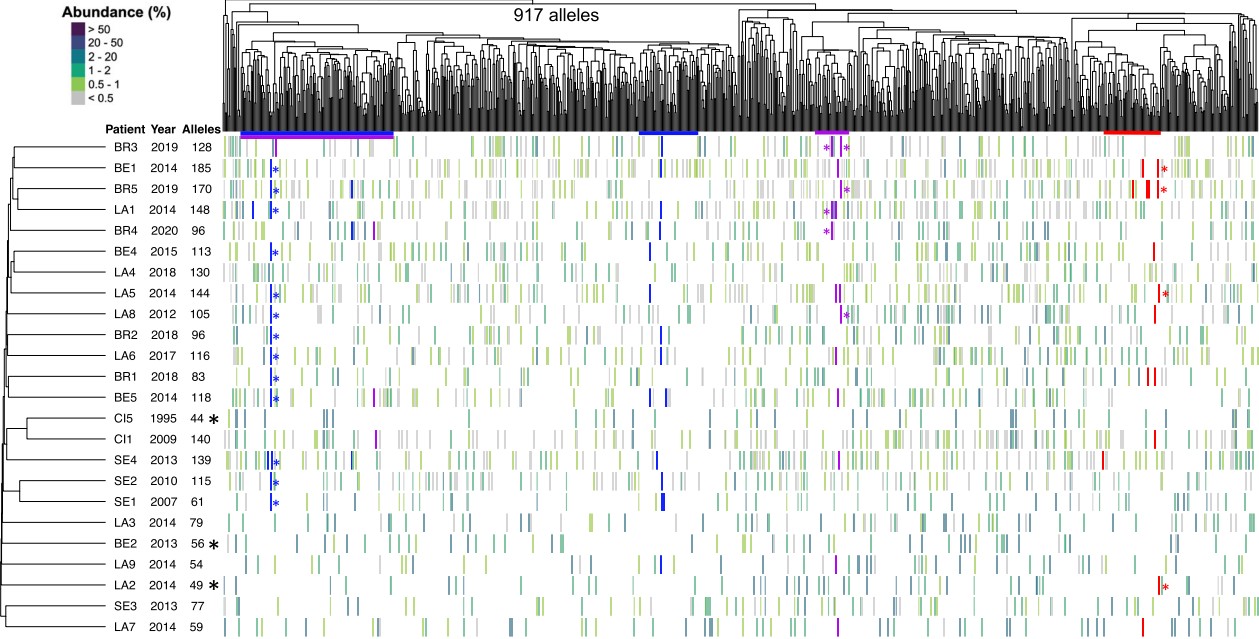

**Fig. 6 | Conservation of fragments within subgroups of alleles among the genomic *msg*-I repertoires of the 24 patients.** Data are added onto Fig. 1. Each vertical line of the heatmap represents an allele present in the given repertoire with the color figuring its abundance in % of all reads composing the repertoire, as indicated at the top left. The black stars identify the three patients assessed to be infected by a single strain (Supplementary Note 2). The presence of each of the three fragments conserved in different alleles is represented by the coloration of the vertical line in blue, purple, or red (red corresponds to the fragment C of 420 bps shown in the alignment of Supplementary Fig. 10, blue and purple correspond to fragments of respectively 133 and 104 bps, each from one of the two other alignments analyzed). The colored stars indicate the two alleles initially aligned in order to identify the duplicated fragments. The colored horizontal lines below the tree show the subgroups of the hierarchical classification tree in which each of the three fragments are present. BE Bern, BR Brest, CI Cincinnati, LA Lausanne, SE Seville. Source data are provided in the Source Data file.

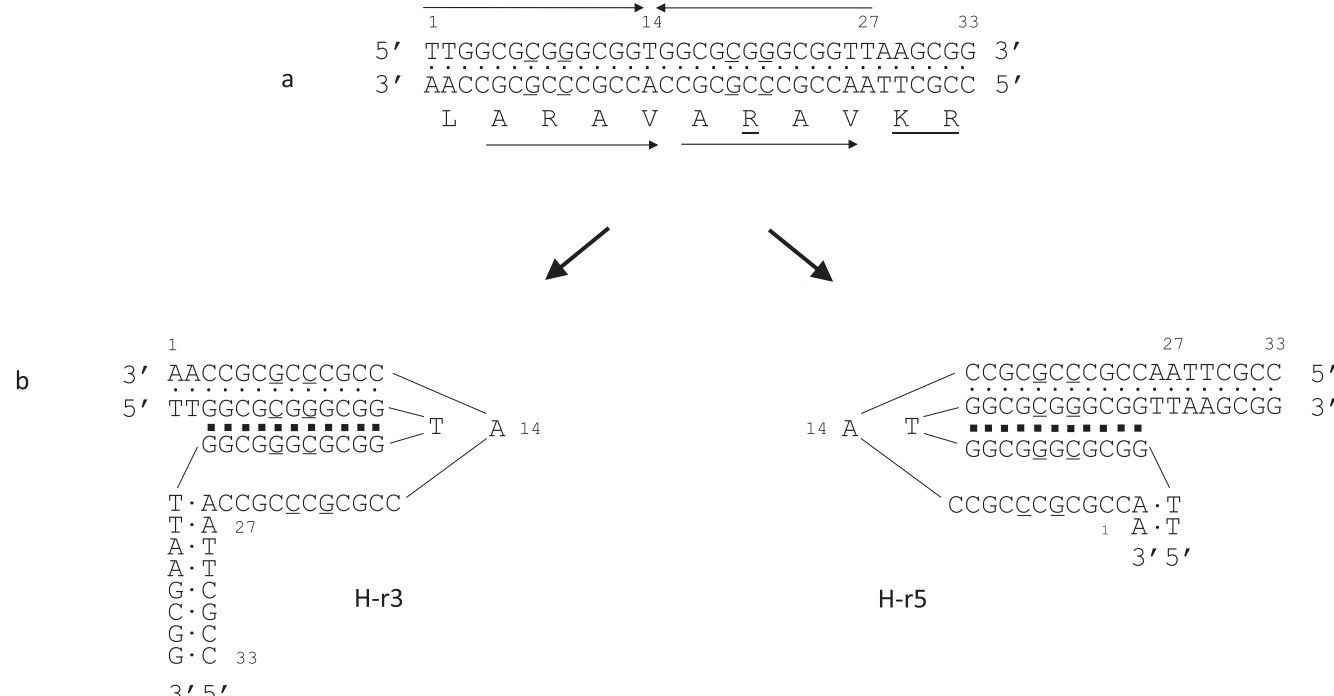

**Fig. 7 | Structure of the conserved Recombination Junction Element (CRJE) and the *H-DNA triplex potentially formed. a** Each strand of the CRJE is enriched in purines or pyrimidines that are part of an imperfect mirror repeats over 27 bps (symbolized by the convergent arrows). The imperfect positions 7, 9, 19 and 21 are underlined. The strand enriched in purines encodes the peptide shown underneath that is part of the Msg-I protein (amino acids' symbols are positioned at the center of the codon). This peptide presents a direct tandem repeat of the motif ARAV (symbolized by arrows pointing to the right). The imperfect C at position 19 of the DNA leads to the underlined R residue in the protein. The repeated ARAV is located before the recognition site KR of the kexin that is underlined (the cleavage by the kexin is believed to remove the constant part of the Msg protein, the UCS, during the maturation[25]). Approximately 11% of the CRJEs presents a transition T to C at position 27 of the DNA, which is a silent polymorphism (Data are provided in the Source Data file). **b** The CRJE can potentially form two isomers of *H-DNA triplex made of base 11 triads involving each 2 or 3 Watson-Crick (points) and 2 Hoogsteen (squares) hydrogen bonds. In the two *H-DNA triplexes, a portion of the strand enriched in pyrimidines is single-stranded over 12 bps. The isomer in which the 5' half of the purines repeat is used as third stand (right, H-r5), rather than 3' (left, H-r3), is commonly less frequently formed[21].

The second new feature of the CRJE is that the peptide encoded presents a direct tandem repeat of the motif ARAV, just upstream of the cutting site of the Kexin that is believed to ensure removal of the constant part corresponding to the UCS[25]. Interestingly, one of the four positions that are imperfect in the repeats, the C at position 19, is necessary to encode the arginine (R) within the second copy of the motif ARAV (Fig. 7a). RA within ARAV is a motif that is cleaved by a number of peptidase, such as the cathepsins (MEROPS database). Furthermore, the R residue is a common recognition site for trypsin-like peptidase. Thus, the CRJE sequences might also be involved in the proper removal of the constant part of the Msg-I proteins by enzymes in order to ensure the variation of each Msg' antigenicity.

As far as the other species of the genus are concerned, *Pneumocystis macacae* and *Pneumocystis oryctolagi* harbor a CRJE sequence similar to that of *P. jirovecii*, including the presence of two R residues in addition to that present in the recognition site of the Kexin (Supplementary Note 7, Supplementary Fig. 13). Thus, these CRJEs might also form a DNA triplex and be recognized by a further protease. On the other hand, nine other *Pneumocystis* species harbor a CRJE more distant from the canonical mirror repeat forming *H-DNA triplexes. However, they might anyway form such structure because the formation of *H-DNA triplex is more versatile and less requiring at the level of sequence than canonical H-DNA[21].

## Discussion

We investigated the mechanisms used by the fungus *P. jirovecii* to vary its antigenicity by analyzing the repertoires of the genes encoding its major surface proteins, as well as the motifs and similarities present within the subtelomeres harboring these genes. Based on our new results, we posit that the antigenic variation system of *P. jirovecii* relies on the three following mechanisms, listed in the order of their importance:

(i) Reassortment of the *msg*-I genes' repertoires and exchange of the expressed allele by translocation of entire genes.

(ii) Rearrangement of the subtelomeres through single recombinations.

(iii) Mosaicism of the *msg* genes through intragenic recombinations.

Although they were all distinct, the genomic repertoires of the *P. jirovecii msg*-I alleles overlapped importantly among the patients. This implies that very frequent translocations of entire non-expressed alleles occur among the subtelomeres, which creates new assortments of the alleles and thus new subtelomeres. The latter would be then segregated into different *P. jirovecii* cell lines, in which further translocations of the *msg*-I alleles would occur. This continuous reassortment of alleles would lead to the distinctiveness of each repertoire that we observed independently of the geographic location, the year of the *Pneumocystis* pneumonia episode, or the subgroups of alleles observed by their hierarchical classification. The likely underlying mechanism of these translocations is a combination of two homologous recombinations, one within the region localized upstream of the *msg*-I CDS and one within the downstream region (Fig. 8a). Such events would permit the exchange of the two alleles (reciprocal exchange), but also possibly the replacement of one by the other (conversion, non-reciprocal exchange). Consistently, the up- and downstream regions flanking the *msg*-I CDSs proved to be similar with

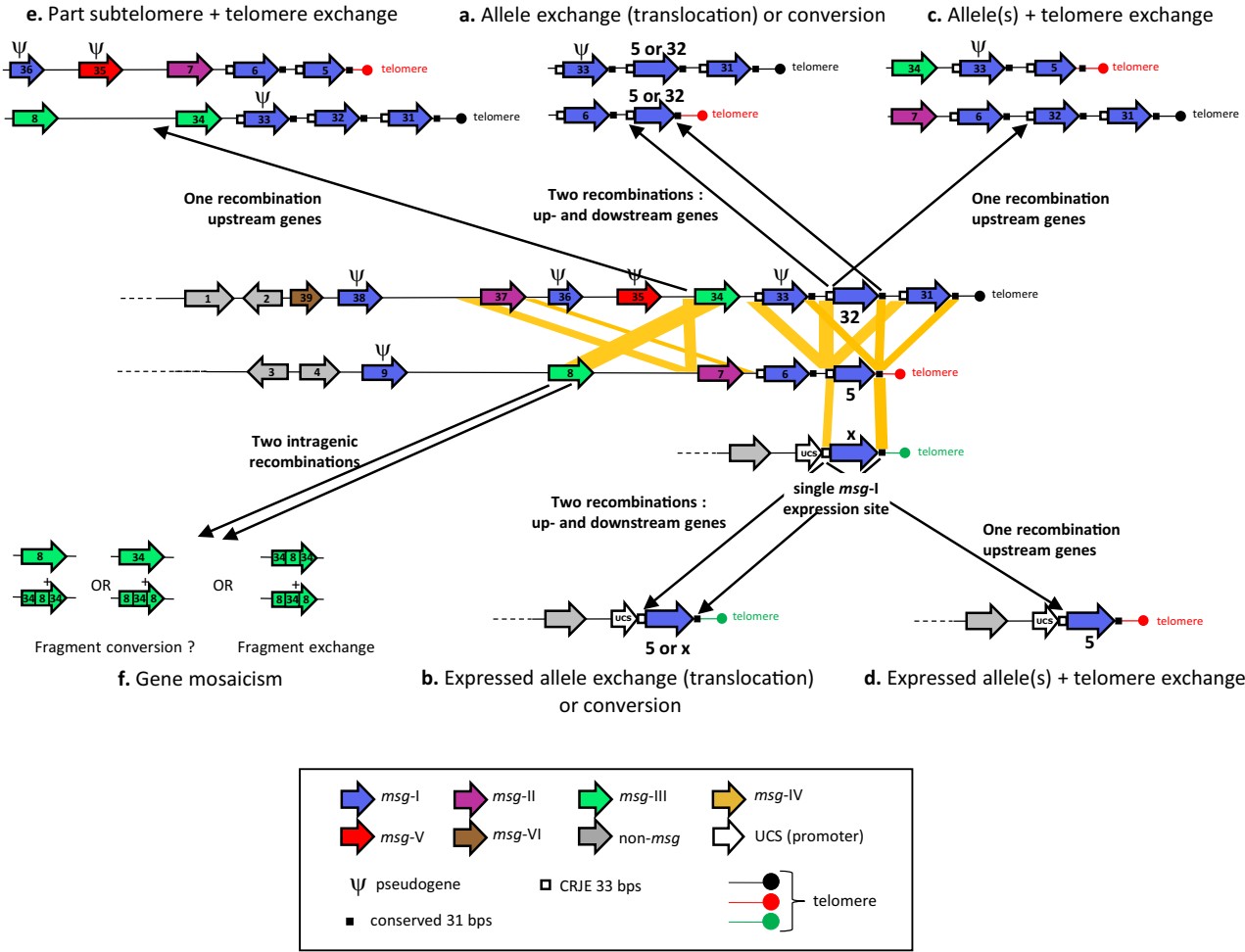

**Fig. 8 | Model for the antigenic variation system of *P. jirovecii*.** The subtelomeres shown are adapted from Fig. 3 of ref. 8 (i.e. contigs 54 and 18). The subtelomere carrying the UCS (upstream conserved sequence) contains the promoter present at a single copy per genome (i.e. contig 72). Relevant similarities between the three subtelomeres are shown by the yellow parallelograms (between up- and downstream regions, and between *msg*-III genes). Recombinations within these similarities lead to the mechanisms indicated by the black arrows. The mechanisms a to f are described in the text. The occurrence of fragment conversions is not supported by the present study (mechanism f). DNA triplexes potentially formed by the CRJE (conserved recombination junction element) sequences might mediate the homologous recombinations.

an identity that was ca. 20% higher than between the genes themselves, probably favouring the postulated recombinations outside of the genes. These recombinations might occur preferentially in the 33 bps CRJE sequence present at the beginning of each CDS and the 31 bps sequence located after the stop codon because these sequences are entirely conserved within all *msg*-I gene sequences reported to date. We also found that the sequences located immediately up- and downstream of the CDSs in *msg* families II, III, IV, and V are also often similar, particularly among the genes located at the tip of the subtelomeres. Furthermore, those of family IV present also an identity higher than the CDSs themselves, like for family I. These observations suggest that translocation of entire genes may also concern these families, but no data supporting this hypothesis are presently available. The facilitation of the translocation of *msg* genes thanks to intergenic spaces more conserved than the flanking genes themselves has been previously postulated in *Pneumocystis carinii*[27].

Translocations through double recombinations are likely to affect also the expressed *msg*-I alleles because they also present the CRJE upstream and the similarity downstream (Fig. 8b). However, most events affect probably the non-expressed alleles because, per genome, the latter are present in ca. 80 copies, whereas there is a single expressed allele per genome. Moreover, translocation of the expressed allele may be reduced by the fact that the similarity upstream of it is only the 33 bps of the CRJE upstream, whereas that of the non-expressed genes is 200 bps-long plus 33 bps of the CRJE. As previously postulated[8,28], an alternative mechanism is suggested by the distal location of the *msg*-I genes within the subtelomeres because it may facilitate their exchange through a single recombination. The latter would occur between two CRJE sequences, leading to the exchange of one or several genes together with the telomere linked to them (Fig. 8c, d).

The number of expressed alleles per patient varied greatly (2–108). Apart from the number of *P. jirovecii* strains infecting the patient, as we evidenced in the present study, one can postulate the four following parameters influencing this number:

(i) The number of cells with distinct *msg*-I repertoires that infected the patient because it can influence the initial number of different expressed alleles.

(ii) The level of immunosuppression of the patient that can eliminate more or less efficiently the cell subpopulations expressing alleles or epitopes previously encountered by the patient.

(iii) The time elapsed between acquisition of the fungus and the *Pneumocystis* pneumonia episode. Indeed, new cell subpopulations expressing a specific allele are presumably segregated continuously and could accumulate over time in immunocompromised patients such as those we analyzed.

(iv) The technical limitations of the methodology may have played a role in the generally low proportion of the genomic repertoires present in the expressed repertoires (<15% in 13 of the 24 patients, Supplementary Table 5). Thereby, the number of alleles in the expressed repertoires may have been underestimated. This might explain that our observations contrast with the 100% of genomic *msg*-I genes being expressed in *P. carinii* and *P. murina* infections[4]. Varying selective pressures by the different hosts may also have been involved.

The conservation of the CRJE sequence *in toto* and in multiple copies in the subtelomeres strongly suggest that it plays a crucial role in the antigenic variation system of *P. jirovecii*. The DNA triplexes that these sequences can potentially form could be involved. However, despite that mirror repeats constitute a hallmark representing up to 1% of eukaryotes' genomes[29–31], their function remains putative so far. This situation results from the fact that DNA triplexes are notoriously difficult to tract[21,23,32]. Nevertheless, a large body of circumstantial evidence suggests that they are involved in a number of genetic processes[33]. Homologous recombination is the process that was most recurrently mentioned because mirror repeats have often been reported close to recombination sites delete ref 28 within the following list of references : [21,33–37]. Furthermore, nucleic acid triplexes (RNA:DNA hybrids, R-loops) would be involved in (i) the switching of the expressed allele participating to the antigenic variation system of *Trypanosoma brucei*[38], and (ii) the switch recombinations in mammalian immunoglobulins[39]. Thus, we hypothesize that the DNA triplexes potentially formed by the CRJE sequences mediate, perhaps activate, the recombinations involved in the system of antigenic variation of *P. jirovecii*.

We found in the present study that the upstream regions of the genes of families II and III are very similar, but not their downstream sequences. Their identity was ca. 30% higher than between the genes nearby (79.8 versus 50.1%). These similarities may promote exchanges of large parts of subtelomeres through a single homologous recombination between them (Fig. 8e). Such intergenic recombinations might be as or more frequent than intragenic recombinations leading to gene mosaicism because the mean identity of the genes within each *msg* family is lower or similar, i.e. 66, 83, 83, 72, 66, 45 for respectively families I–VI[8]. This mechanism would lead to the creation of new subtelomeres that are potentially of varying sizes. This would be compatible with the important variation of the telomere length that we observed[8] (Supplementary Figs. 7 and 8). Indeed, the distance between the genomic genes and the *msg*-I genes with a CRJE sequence, which are the genes always closest to the telomere, varied from 6 kb (contig/subtelomere 149, Supplementary Fig. 8a) to 25 kb (54, Supplementary Fig. 7).

A general phenomenon concerning subtelomeric genes is the occurrence of ectopic recombinations between them, i.e. also between non-homologous chromosomes[16,40]. These recombinations may occur mostly when the telomeres and subtelomeres are bundled as a "bouquet" by the attachment to the spindle body during the prophase of meiosis[16,41,42]. The products of two such recombinations are difficult to assess, and may vary according to the organism (Fig. 8f). Moreover, a balance between fragment conversions and fragment exchanges may exist in all organisms because the former generally homogenizes the alleles, whereas the latter diversifies them[43]. The importance of each process in the balance may depend on the amount of different alleles imported in the system, for example by mating of strains because it leads to the fusion of two sets of alleles. The recombinations responsible for the mosaicism of the alleles could also result from the meiotic reparation of the altered genes through fragment conversions[44].

Ectopic recombinations are likely to generate the mosaicism of the *P. jirovecii* genes, pseudogenes, and intergenic spaces of the subtelomeres that is observed. However, our present observations suggest that the duplicated fragments we observed among the *P. jirovecii msg*-I alleles have been conserved from ancestral alleles during the diversification of the alleles. Indeed, two thirds (4 out of 6) of the fragments <200 bps that we investigated were present in two subgroups of distant alleles. One was even in the two main subgroups of *msg*-I alleles, i.e. very distant ones. On the other hand, all ten fragments ≥200 bps were present in a single subgroup. During the diversification process, the probability for a fragment to be split is probably greater with a length ≥200 than for those <200. Thus, the larger fragments would more likely be present within a single subgroup than the shorter ones.

An alternative hypothesis is that the duplicated fragments result from conversion events that occurred within subgroups of related alleles, followed by the dissemination of the alleles created among patients. The presence of the same duplicated fragments within two distant subgroups, as we observed, could be explained by the existence of hotspots of recombinations along the gene leading to the same duplicated fragments. However, our analyses of the locations of the duplicated fragments did not reveal the presence of any such hotspots. Therefore, our observations favor the hypothesis that the diversification of the *msg*-I alleles in *P. jirovecii* results primarily from recombinations that split ancestral alleles. These recombinations could be those resulting in fragment exchanges, events we could not detect by our approach, or could be the single ones resulting in the rearrangements of the subtelomeres mentioned here above. This conclusion contrasts with those of previous studies. Indeed, conversion events were suggested by the high content of G + C motifs within *P. jirovecii msg*-I genes[45], and by the analysis of duplicated fragments at the *P. carinii* expression site[46].

Many different *P. jirovecii msg*-I alleles were observed being expressed in the lungs of the immunosuppressed patients analyzed here. In contrast, immunocompetent individuals that are colonized by the fungus might harbor a smaller number of expressed alleles because of their effective immune system, but no data are presently available. The latter immunocompetent individuals constitute probably the niche in which the antigenic variation system of *P. jirovecii* evolved, so that this system is above all a colonization factor. Thus, to improve our understanding of the latter, it might be useful investigating the immunocompetent transitory carriers such as healthcare workers in contact with patients with *Pneumocystis* pneumonia, or infants experiencing their primo-infection by *P. jirovecii*.

The important overlap of the *msg*-I genes' repertoires implies the existence of a very efficient mean of dissemination of the alleles and strains by frequent contacts between the *P. jirovecii* populations. This might happen through the primo-infections of infants that are frequent events occurring in the general population. Another not mutually exclusive hypothesis is the transient carriage of the fungus by healthy people, a phenomenon that has been frequently hypothesized but not firmly established so far[47–49].

Our data support that the strategy of *P. jirovecii* is the continuous production of new subpopulations that are antigenically distinct, as we previously proposed[8]. This strategy relies on recombinations but their frequency and the speed of the subtelomeres evolution over time remain to be determined, possibly by the analysis of sequential samples from the same patients. The analysis of such samples may also allow tackling a crucial aspect of immune evasion that our study has not investigated: the change of the Msg glycoproteins at the cell surface overtime. Surface antigenic variation at each generation being probably necessary for survival, it might account for the obligate sexuality of this fungus[50,51]. Consistently, the latter probably ensures also proliferation of the fungus[50,51]. The strategy of *P. jirovecii* is one of a kind among human pathogens and might be adapted to the nonsterile niche constituted by the mammal lungs[8,52]. It contrasts with the populations that are

antigenically homogenous of *Plasmodium* and *Trypanosoma*, and that vary overtime upon exchange of the expressed gene[53]. These might be imposed by sterile niches such as blood.

# Methods

## Ethics approval and consent to participate

In Lausanne, Bern, and Seville, the procedure for admittance in the hospital included informed written consent for all patients. The admittance form included the possibility to require their samples not to be used for research. The samples were obtained through the hospital's routine procedure and were anonymized. The research protocol was approved by the Seville Hospital review board and the Swiss institutional review board (Commission Cantonale d'Éthique de la Recherche sur l'Être Humain, http://www.swissethics.ch). The collection and use of archival specimens was approved by the ethics committee of Brest University Hospital on June 24th 2021, and registered with the French Ministry of Research and the Agence Régionale de l'Hospitalization, No. DC-2008-214. The samples from Cincinnati were anonymized; their use was not for research on human subjects and did not require approval by the Institutional Review Board.

## Samples and DNA extraction

Broncho-alveolar lavage (BAL) samples were collected from 24 immunocompromised patients with *Pneumocystis* pneumonia in five different geographical locations over 25 years (Supplementary Table 1). DNA was extracted from 0.2 ml of each BAL using the QIAamp DNA Blood Mini Kit (Qiagen).

## Amplification of the repertoires of *msg*-I genes

Two distinct generic PCRs were used to amplify all entire *msg*-I genes that are expressed (the "expressed repertoire"), or all entire *msg*-I genes of the genome that are present in the sample, i.e. the expressed plus the non-expressed genes (the "genomic repertoire"). The expressed repertoire was specifically amplified thanks to a forward primer localized at the end of the UCS sequence, 27 bps upstream of the CRJE (GK135: 5' GACAAGGATGTTGCTTTTGAT 3')[54]. The genomic repertoire was amplified specifically using a forward primer covering the 3' half of the CRJE sequence (CRJE-for-bis: 5' TGGCGCGGGCGGTYAAG 3'; the underlined Y was introduced because ca. 89 and 11% of the CRJEs harbor respectively T and C at this position; Source data are provided in the Source Data file). The reverse primer was identical for both PCRs and located in a conserved region of 31 bps ca. 90 bps after the stop codon of the *msg*-I genes (GK452: 5' AATG-CACTTTCMATTGATGCT 3'; the underlined M was introduced because ca. 90 and 10% of the sequences harbor respectively T and G at this position; this primer is identical to that previously published[20], including the wobble). The alignment of the 48 published sequences including the region is shown in Supplementary Fig. 1, only one might be less efficiently amplified because of the presence of a G instead of a A at the 3' end of the primer. PCR was performed with one microliter of DNA from the BAL in a final volume of 20 μl containing 0.2 μl polymerase (KAPA LongRange HotStart, Roche), the provided buffer, each dNTP at 0.2 mM, each primer at 0.5 μM, and a final MgCl$_2$ concentration of 3 mM. In order to prevent contaminations, PCRs were set up and analyzed in physically separate rooms using materials dedicated to each room, and negative controls were systematically carried out at each experiment.

A touchdown PCR procedure was used for both PCRs. To amplify the expressed repertoire, the program began with 10 cycles consisting in a constant decrease of the annealing temperature from 62 °C to 55 °C, followed by 25 or, if needed to obtain a sufficient amount of PCR product, 30 cycles with annealing at 55 °C. For the genomic repertoire, the decrease was from 68 °C to 58 °C in 10 cycles, followed by 20 or 25 cycles with annealing at 58 °C. The elongation time was for both PCRs 3 min at 72 °C. The reactions finished with a final elongation of 7 min at

72 °C. The PCR products were verified by (i) their size of ca. 3100 bps on an agarose gel and (ii) subcloning some of them using the TOPO cloning kit (Invitrogen) followed by Sanger sequencing. The PCR products were then purified with E-gel CloneWell 0.8% (Invitrogen) according to the manufacturer's instructions, except for the use of 50% glycerol instead of the provided loading buffer to avoid any issues during the sequencing due to its unknown composition. Finally, they were evaporated on a heating block at 50 °C with the tube lid open to reach the adequate concentration required for PacBio circular consensus sequencing. Each open tube was covered with a non-airtight lid during evaporation in order to prevent cross-contamination. Absence of the latter was assessed by the presence of a single allele in the plasmid controls upon sequencing. Several PCR products from 38 out of the 48 samples with less than 1.0E + 07 *P. jirovecii* genomes per ml had to be pooled after the E-gel step to obtain enough DNA for PacBio sequencing.

## PacBio circular consensus sequencing (CCS)

Single molecule real time sequencing of the PCR products was performed using the PacBio Sequel II at the Genomic Technologies Facility of the University of Lausanne, Switzerland. Barcodes were added to the amplicons before pooling them. They were then circularized to enable multiple sequencing of the same *msg*-I gene and the generation of circular consensus sequences. This technology provides long and accurate reads of the repetitive sequences present in the *msg*-I genes. The sequencing of 50 samples (48 PCR products from the 24 samples, plus two plasmid controls) resulted in 10'223 to 160'836 reads per sample.

## PCR artefacts

The production of artefacts during PCR amplification of mixed alleles results in additional non-biological diversity[55]. In order to address the occurrence of this problem in our conditions, two plasmids containing each a single specific *msg*-I allele were mixed before or after PCR amplification, and then sequenced using PacBio CCS (plasmids PL1 and PL2 containing alleles PL1c172754267 and PL1u100008324 that show 62% identity, which is close to the average of 66% between all 1007 alleles observed in the study [Supplementary Data 5]). Chimeric amplicons were observed upon mixing before PCR, but not after PCR. Most of the reads corresponded to the two alleles present in the plasmids (92.5%), 7.5% presented a single nucleotide polymorphism probably corresponding to a PacBio error, and 0.33% were chimeras explainable by a single recombination in the first or last 500 bps. These latter PCR artefacts result probably from a premature detaching of the DNA polymerase during the elongation step that produces incomplete strands. The latter can then be used as primers by annealing to closely related sequences. The putative extremities of the incomplete strands were localized in the first and last 500 bps of the chimeric sequences. These findings led to the decision to trim both ends of the ca. 3.1 kb sequences and to keep only the central 2 kb for the subsequent analyses (positions 500 to 2500). This trimming reduced the diversity of the alleles identified by ignoring some variation but was necessary to avoid the artefactual chimeras. It affected marginally the results presented and had no impact on the conclusions drawn.

## Allele identification and quantification

A dedicated bioinformatics pipeline of analysis of the Pacbio CCS raw reads was developed to identify the alleles present in each sample as well as their abundance. Information about the versions of the software and its packages are provided in Supplementary Table 2. The multistep process consisted in the following sequence of analyses:

1. hmmer[56] was used to identify the reads corresponding to *msg*-I genes by aligning them to a profile-HMM specifically generated using 18 *msg*-I genes previously identified in a pilot experiment and trimmed after the CRJE in order to be in frame. The nhmmer

program command was used to filter the reads by alignment length and bit score, which resulted in removal of ca. 5% of the reads and a final number of 9742 to 156,375 reads per sample for further analysis.

2. swarm (version 3.1.0)[57] was used to define cluster seeds for each sample. Each read that is more than 1 bp (option -d 1) different from any other is defined as a new cluster seed by swarm. The seeds with more than one read in their cluster were kept as cluster seeds for the following clustering step.

3. cd-hit (version 4.8.1)[58] was used to cluster the reads around the cluster seeds defined by swarm. The command cd-hit-est-2d, with options -c 0.99 and -g 1, allowed the allocation of each read to the most similar seed to create clusters of similar reads and determine the abundance of the alleles that were identified among the reads as described in the next section.

The fine-grained clustering obtained so far still encompasses PacBio sequencing errors and an additional step of clustering was required to get rid of this problem. Two plasmids containing each a single specific *msg*-I allele, one with the UCS and the other without the UCS, were amplified by PCR, PacBio CCS sequenced, and clustered as described above, yielding several seed sequences. However, all pair-wise sequence identities between the seed sequences from a single plasmid were higher than 99.5%, which matches the expected error rate of the PacBio CCS sequencing[59]. An additional round of clustering was performed with an identity threshold set at 99.5%. This yielded clusters of reads that were considered as identical alleles, or alleles that cannot be distinguished by the sequencing technique used. This corresponds to a difference of less than 10 bps in the trimmed 2 kb sequences.

### Confirmation of the *msg*-I alleles repertoires
The repertoires observed were supported using PCRs specific to three given alleles among five patients (Supplementary Table 3, Supplementary Fig. 2). Three alleles were selected that were present in several patients and absent in others. Primers were designed within these alleles to amplify each of them specifically (Supplementary Table 3). The specificity of the primer pairs were assessed by blasting them against the 1007 different *msg*-I alleles identified in the genomic plus expressed repertoires observed in the present work, as well as against the whole nucleotide collection (nr/nt). The PCRs were performed with the following parameters: 3 min at 94 °C followed by 35 cycles consisting of 15 s at 94 °C, 30 s at 56 °C, and 2 min at 72 °C, followed by a final extension of 7 min at 72 °C. The presence or absence in patients observed by PacBio CCS was confirmed by the specific PCRs (Supplementary Fig. 2). The sequences of the amplicons obtained using the Sanger technology showed 100% identity with those obtained using CCS. The positive DNA control was produced by random amplification of 2.5 μl DNA from the BAL of patient LA2 using the GenomiPhi HY kit (GE Healthcare), followed by a purification step using the columns of the QIAamp DNA Blood Mini Kit (Qiagen). It was also used for the set up of the PCR reactions.

### Estimation of the number of *P. jirovecii* strains
For the sake of clarity, we use in the present manuscript the term "strain" despite that the correspondence of each genotype identified to a biological strain cannot be ascertained in absence of culture in vitro. The number of *P. jirovecii* strains infecting each patient was estimated by amplifying and sequencing the region comprising the internal transcribed spacers and the 5.8S rRNA gene of the ribosomal RNA operon (ITS1-5.8S-ITS2). The PCRs mixes were identical as for the amplification of the *msg*-I genes with one microliter DNA in a total volume of 20 μl. The PCRs were performed with the following parameters: 3 min at 95 °C initial denaturation followed by 35 cycles consisting of 30 s at 95 °C, 30 s at 62 °C and 45 s at 72 °C followed by a final

elongation of 1 min at 72 °C. The primers were derived from those described by reference[60]: 5′ GCTGGAAAGTTGATCAAATTTGGTC 3′ and 5′ TTCGGACGAGACTACTCGCC 3′ (the six underlined bases were added in 5′ to the first primer to adjust the annealing temperature, and the four underlined bases replaced those from *P. carinii*, the species infecting rats, that were in fact within the second primer). The PCR program included 3 min at 95 °C of initial denaturation, 30 to 35 cycles consisting of 30 sec at 95 °C, 30 s at 62 °C, and 45 sec at 72 °C, followed by 1 min of final elongation at 72 °C. After purification using the Qiagen PCR column kit, the amplicons were sequenced using PacBio CCS. The bioinformatic pipeline dedicated to the analysis of the ITS1-5.8S-ITS2 reads started with hmmer to identify the correct genes followed by a filtering step by read size between 470 and 500 bps (see above, Allele identification and quantification). Xue et al.[60]. Highlighted that the ITS1-5.8S-ITS2 region includes seven homopolymer stretches that are prone to errors during both amplification and sequencing. Hence, the length of the six homopolymers that showed variation in our data were homogenized (Supplementary Fig. 3). The redundant reads were then clustered using swarm (option d0) and only the clusters comprising more than 1% of the reads present in each patient were analyzed. The correspondence of the sequences of the 25 genotypes observed to those present in GenBank is given in Supplementary Table 4.

### BLASTn analyses
We searched similarities within the 37 subtelomeres present in the whole *P. jirovecii* genome assembled from a single strain using PacBio sequencing[8]. We used the BLASTn algorithm on the NCBI website (https://blast.ncbi.nlm.nih.gov/Blast.cgi) with the default parameter values (expected threshold 0.05) and the following settings: database: whole-genome shotgun contigs (wgs), limit by: BioProject, BioProject name: 382815 *Pneumocystis jirovecii* strain E2178, program selection: somewhat similar sequences (BLASTn). The 37 contigs carrying *msg* genes are entire or partial subtelomeres corresponding to most of the 34–40 that are present in *P. jirovecii* genome (17–20 chromosomes)[4]. Nevertheless, 17 of the 37 contigs do not carry non-*msg* genes flanking the subtelomere, so that they could not be attributed to a specific chromosome previously described[4]. Consequently, some of them could be parts of the same subtelomere, rather than distinct ones. For simplification, each contig is considered as one specific subtelomere in the present manuscript.

### Search of mosaicism within the *msg*-I alleles
Duplicated fragments were used to search mosaicism within the 917 alleles identified in the genomic repertoires of this paper. An allele chosen randomly was blasted using BLASTn with default parameters to align two or more sequences among the 3.1 kb sequences of the 917 alleles (the full size alleles were used because search of duplicated fragments is not impaired by eventual chimera). Its closest allele, i.e. that with the highest score, was chosen for the following steps. Alignment of both sequences allowed the visual identification of fragments larger than 100 bps shared by the two alleles. These fragments were then searched among the 917 alleles using BLASTn, identifying the ones containing the fragments completely conserved.

### Bioinformatics search for site-specific recombinases
Potential genes encoding site-specific recombinases were searched for in the *P. jirovecii* genome (accession number LFWA01000000) by matching this genome against large pools of representative bait sequences by using tBLASTn (NCBI BLAST suite). These pools were recruited through the InterPro annotations IPR011010 (DNA breaking-rejoining enzyme, catalytic core) from a wide range of taxa. The Uni-Prot identifiers of the baits used were P03870, P13769, P13770, P13783, P13784, and P13785. To avoid missing candidates because of the use of too stringent conditions, the tBLASTn searches were conducted with relaxed parameters (E value from 1E-4 down to the default value). Each

match with a suitable E value was investigated by looking for pre-existing annotations. If no coding sequence (CDS) annotations were available, the matched region was assessed for putative novel CDSs and their translated sequence were submitted to the InterProScan4 tool to detect the required reference signature matches[26].

### Reporting summary
Further information on research design is available in the Nature Portfolio Reporting Summary linked to this article.

### Data availability
The PacBio CCS raw reads generated in this study have been deposited in the NCBI Sequence Read Archive and are accessible through accession code SRP434110. The *msg*-I raw data are accessible through accession numbers SRR24284242 to SRR2428430 and the ITS1-5.8S-ITS2 raw data under SSR25739987 to SRR25740015 in BioProject accession number PRJNA936793. The sequences obtained in this study have been deposited in Genbank (1007 new *msg*-I alleles) under accession numbers OR489167 to OR490173 and 15 new ITS1-5.8S-ITS2 alleles under accession numbers OR475686 to OR475700. A table including the relative abundance of each *msg*-I allele identified in each patient is provided as Supplementary Data 5. Source data are provided with this paper.

### Code availability
Computer codes used to generate results are deposited at https://doi.org/10.5281/zenodo.8363954[61].

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

## Acknowledgements

The present work was submitted by C.S.M. as partial fulfillment of a Ph.D. degree at the Faculty of Biology and Medicine of the University of Lausanne. Sequencing was performed at the Lausanne Genomic Technologies Facility, University of Lausanne. We thank Patrick Taffé (Unisanté, Division of Biostatistics, University of Lausanne) for input into the simulation experiments. This work was supported by Swiss National Science Foundation grant 310030_192802 to P.M.H. The Swiss National Science Foundation had no role in any steps of the study. M.T.C. is a Senior Research Career Scientist supported by IK6BX005232 Department of Veterans Affairs. All materials are available from the corresponding author.

## Author contributions

P.M.H. designed the study. C.S.M. and S.R. performed lab experiments. C.S.M. performed bioinformatics analyses under the supervision of M.P. JMGCFA performed bioinformatics searches. K.M., G.N., M.T.C., and E.J.C. prepared the samples and were involved in the writing of the manuscript. P.M.H. and C.S.M. wrote the first draft of the manuscript. All authors reviewed the manuscript.

## Competing interests

The authors declare no competing interests.
