## [Peer Review File · Nature Communications]

Fungal antigenic variation using mosaicism and reassortment of subtelomeric genes' repertoiresREVIEWER COMMENTS

Reviewer #1 (Remarks to the Author):

The manuscript describes the repertoires of *Pneumocystis jirovecii* major surface glycoproteins sequenced after PCR amplification from clinical samples. The findings essentially corroborate previous observations of msg expression and lead the authors to make some interesting hypotheses regarding molecular mechanism of antigenic variation in this fungus. I think, in terms of conclusions that may be sustained by the data, the findings represent only a modest advance (i.e. principally, how repertoires from different clinical samples compare) on what was already known (i.e. that there are distinct msg subfamilies with specific and fluid genomic dispositions, monoallelic expression, mosaic sequences etc). The comparison of clinical samples seems not to provide explanation for repertoire (in terms of sample provenance), but is more consistent across samples than would be the case for other antigenic variation systems. The novelty in the manuscript comes from the new ideas around molecular mechanism (i.e. subtelomeric reorganisation, the involvement of DNA triplexes). While plausible and attractive, these are not tested, and seem to derive from published genomic data not derived from the clinical samples. Overall, I think the work could go much further to address its own ideas, even without an in vitro culture system. Relying on sequencing of PCR products seems to me to be unnecessarily limited, and lacking internal corroboration by other methods. I suggest some extensions below that I do not think are beyond the scope of the questions posed, and which would represent significant advances of what we already know.

Major points:

1. Why is there no RNAseq of msg mRNA? Are the authors assuming that 'expressed' genes are exclusively associated with the previously observed expression site? (p19). Can they guarantee this? It should at least be confirmed independently with transcript sequences.
2. Ideally, this would be single-cell RNAseq, providing much richer data on cellular heterogeneity in msg expression and directly quantifying the subpopulations the authors predict to exist. I see no reason why this should not be possible to develop from the starting material.
3. Why is there no genome sequencing to provide new data on the dynamism of subtelomeric loci? Comparison of clinical samples would then confirm the subtelomeric recombination events that the authors imagine, as well as define the recombination breakpoints (p26), perhaps giving a novel mechanistic insight. This is possible from clinical samples, using cell sorting and single-cell sequencing. There are several commercial products to enrich sequences for known targets, such as *Pneumocystis* msg loci.
4. Why is there no formal analysis of recombination (i.e. some kind of phylogenetic profiling, GARD, TOPALi, PHI-Pack etc.) to distinguish mosaicism from alternative explanations such as independent sorting of divergent gene duplicates (time series RNAseq would also provide a critical perspective on this). I wasn't sure how the authors defined a mosaic here, but it seemed to involve BLAST only.
5. Why not follow up the smaller group of msg-I that occupy expression-sites in all patients: are they surface expressed? Presumably you can raise an antibody to them, and obtain *Pneumocystis* from clinical samples by cell sorting?
6. Why is there no serology? I would imagine that host antibodies are present in same clinical samples? Testing the serological response to observed msg using simple methods would test the idea that rarer alleles are less immunogenic (such frequency dependency being critical to antigenic variation).

Minor points:

1. Please use line numbers
2. P2, l13: "The recombination also generates new mosaic genes constantly"
3. P3, l15: remove "it"
4. P3, l22-3: "recombination of genes in this genomic region is advantageous as it has no, or little, effect..."
5. P4, l1: replace "single" with "one gene"

6. P4, l5 "...is also found at the start of each msg-I allele"
7. P9, l6: should this sentence read "so far DOES NOT account for..."?
8. P9, l15: "...sequencing technique used."
9. P9, l17: "...present IN patients..."
10. P9, l22: "...amplify each of them specifically (Table S3)."
11. P11, l12: "BLASTn analyses"
12. P12, l4: "Duplicated fragments were used to search..."
13. P12, paragraph 2: I am unsure from the text what exactly the purpose of this activity is. What exactly were you looking for in your sequences. How did you define what was a mosaic?
14. P15, l11: what is meant by "related ones"? A better expression would be "There were no clades of msg-I sequences in our trees that could be associated with a specific repertoire, suggesting that sample provenance does not explain the tree topology."
15. P16, l8: 'Conversely', not "reversely".
16. P18, l5: 'Consistent' not "consistently"
17. P18, paragraph 2: I think the use of homo/heterogeneous here is not quite correct. A situation where most abundance belongs to just a few alleles would seem to be more homogeneous than a situation where distribution is more equitable, but you seem to be suggesting the opposite. In any case, I think you mean that abundance has an overdispersed distribution (most sequences come from a few types) in expressed repertoires, and a more normal/underdispersed distribution in complete repertoires.
18. P22, l3: "To understand the phenomenon better,"
19. P22, l10: "The 33bp CRJE sequence..."
20. P32: Plasmodium

Reviewer #2 (Remarks to the Author):

The authors have evaluated the diversity of the Msg-I gene family repertoire in *Pneumocystis jirovecii* and explored the level of conservation among the other Msg gene families. They also describe hypothetical mechanisms for recombination in the Msg-I gene family. This is the most extensive and detailed analysis of the Msg-1 repertoire among patient isolates conducted to date, and helps clarify the broad diversity of the repertoire, the geographic distribution of individual alleles, and the importance of recombination in generating Msg-I diversity.

While the study provides important insights into the organization of the Msg genes, there are a number of issues to consider that can potentially improve the manuscript.

Major comments:

1. A major focus of the paper is to characterize the Msg-I repertoires of *P. jirovecii* strains. However, this analysis is greatly complicated by the fact that the majority of patients were infected by multiple strains, making it difficult to describe the repertoires in single strain/organism. Particularly, with the methods presented, it is impossible to determine if the msg sequences identified in these patients (with multi-strain coinfection) were from the same or different strains. This caveat, coupled with the fact only a subset of the Msg-I repertoires were obtained from the majority of patients, limits the depth of insights into the population structure, diversity and potential recombination patterns of the msg-I genes.
2. The authors suggest that all the Msg genes in the *Pneumocystis* genome are being amplified by the genomic PCR (the complete repertoire), but their own data suggests that this is not the case for all samples. If the entire repertoire is being captured in each PCR reaction, replicate amplifications should be identical, but as shown in Table S5, this is not the case for either the complete or the expressed repertoire. While low abundance may explain the discrepancy in the expressed repertoire, this should

not be the case for the entire repertoire, especially those samples with a single strain, as each *Msg* gene should be present in equal numbers, as a single copy per genome, with an expected ~80 copies/genome. In samples with more than one strain, there should be an equal abundance of the ~80 *Msg* genes for each strain, although the strains may be present in varying proportions. Assuming that each organism has a similar number of *Msg* genes per genome, It is surprising that many samples have substantially fewer alleles identified.

Moreover, as the authors note, all the expressed *Msgs* should be identified in the complete repertoire, but again this is not the case. The authors again suggest that this is due to low abundant alleles, but as above, this does not seem to explain the incompleteness of the genomic repertoire.

PCR for both the complete and the expressed repertoires relies on a previously reported primer which is located in the 3' UTR of the *Msg-I* genes. While the primer sequences appear to be well conserved, do the authors have evidence supporting that this 3' UTR region is linked to all *Msg-I* genes and is conserved among different *P. jirovecii* strains, i.e. that it can in fact be used to amplify the entire repertoire among all isolates? It would be helpful to note that an alignment of this primer with all *msg* genes containing the 3' UTR (including the primer binding region) that were not PCR generated verified this. Whether these primers are able to amplify only partial or all *Msg* genes could also be examined by Illumina NGS of total genomic DNA (rather than a PCR product) to look for polymorphisms in the regions of the primers. Even if the primers are not fully conserved, the study still provides important insights into the subset of *Msg-1* genes that can be amplified.

Furthermore, the efficiency of PCR to amplify the *Msg-I* repertoires could be greatly affected by the *P. jirovecii* DNA concentrations in clinical samples. To avoid or reduce the possibility of a partial or biased amplification, it would be helpful to determine the minimal *Pneumocystis* DNA concentration (ideally by qPCR to quantify the *P. jirovecii* genome or *msg* copy number) in clinical samples that would allow an amplification of the entire *msg-I* repertoires.

3. The authors have trimmed ~1,000 bp (~500 bp at either end) prior to conducting their analysis, to minimize inclusion of artificial chimeric *Msgs* in their analysis. However, the trimming was not made at the same positions for all *Msg* sequences, presumably due to the presence of indels. For example, for the *Msg* sequences from patients LA1, LA3 and BE1, the 5' ends of the retained sequences had 33-55bp size differences between the shortest and longest ones while the 3' ends of the retained sequences had 148-165bp size differences between the shortest and longest ones. It would be beneficial to clarify the reasoning behind the uneven trimming and explain how this was factored into downstream analysis. A drawback of this trimming approach (1/3 of the full length) is that it may result in a further reduction in the number of unique alleles; alleles that appear identical may in fact not be identical since there can be substantial variation in the trimmed sequences, especially in the first ~500 bp. This should be taken into account when they draw their conclusions.

4. On page 23 the authors speculate that the RA in the CRJE peptide may serve as a target for host peptidases to remove the constant part of *Msg* proteins to maximize antigenic variation. The authors, however, provide no data to support this hypothesis, and it seems questionable for a couple of reasons. First, the RA site is not conserved across *Pneumocystis* species. Given that the *Msg-I* gene family system has been conserved across all *Pneumocystis* species examined to date, and thus presumably dates back to a primordial ancestor of all *Pneumocystis* species, it's likely (though not certain) that the mechanisms for ensuring antigenic variation are similar across species. Moreover, if host proteases are processing, then host immune cells would have access to the fragments as well and could potentially develop an immune response to the fragments, potentially to the detriment of the organism. Experimental data need to be provided to support this hypothesis.

5. Similarly, on page 22 the authors hypothesize that the CRJE region contains DNA elements that allow formation of a DNA triplex, and suggest that this triplex may play a role in translocation of entire *Msg* genes. They note however, that the CRJE of *P. murina* is unlikely to form such a triplex, and that of *P. carinii* could possibly form one, though it has a less symmetrical mirror repeat. Again, given that the *Msg* system for presumed antigenic variation has been maintained across *Pneumocystis* species

since its origin in a common ancestor, it seems likely that a mechanism for recombination would also have been maintained. It would again be important to have experimental support of this hypothesis to include it in the manuscript. The current conclusion in the discussion that "Reassortment of the msg-I genes' repertoires and exchange of the expressed allele by translocation of entire genes mediated by DNA triplexes" is not warranted specifically as related to DNA triplexes.

6. There are inconsistencies in the number of alleles identified in the study. Page 12 noted that 917 alleles were identified in the complete repertoires, while the last line on page 9 notes that 1007 alleles were identified. The authors note on page 15 and in Table S4 that the complete repertoire contained 44 to 185 alleles, and the expressed repertoires contained 2 to 108 alleles, but later note, line 388 and Table 1, that one sample had 411 unique alleles in the complete repertoire, and similarly that the same sample had 347 expressed alleles. However, none of the other tables or figures (e.g. Figure 1, Table S4) include a sample with that many alleles. Figure 4A and 4B similarly noted that e.g. 88% and 88.5% of the alleles in single cities were present in a single patient sample, but the graphs do not seem to identify a patient with that predominance of alleles. This inconsistency needs to be clarified. For the sample with 411 unique alleles, how many *Pneumocystis* strains were identified in the sample? Table S4 also lists two patients infected with a single strain (LA8 and SE2); each of them had 105 or 115 alleles, which are substantially more than other samples and the expected total number of ~80 per strain. Although it is likely that the total number of alleles varies among different strains, given that these are substantially more than anticipated in a single strain, it would be important to rule out the possibility of inappropriate classification of those alleles. Are the sequence divergence levels among those alleles in each sample significantly lower than those in other samples or are there any unique sequence patterns in those samples? Another possibility is that these represent coinfection with multiple strains which were indistinguishable by the genotyping method used. It may be worth genotyping using additional markers with high discriminatory power. It would be helpful to add GenBank numbers to Table S4 for all genotypes identified.

7. It would be helpful to know the organism load in each sample, based on qPCR, if that is available. Normalizing the *P. jirovecii* concentration in all samples or at least those with high organism burdens may help reduce the variation in the number of alleles though this may be challenging due to the presence of coinfection with multiple strains. Did the samples with a low expressed repertoire (e.g. BR1, BR2, and BR3) have a low overall organism burden? If so, that might explain the low expressed repertoire. If not, is there anything unique about the patients from whom these samples were obtained? It's noteworthy that they represent at-risk patients with 3 different underlying diseases (giant cell arteritis with presumed corticosteroid therapy, cancer, and HIV infection).

8. While it's clear that mosaicism plays a role in generation of Msg diversity, the time-line for this is unknown; the conclusion that there is constant generation of new mosaic genes (in the Abstract) is hypothetical but currently not supported by experimental evidence. Despite the great variability of the msg-I repertoires among unrelated *P. jirovecii* isolates, there has been no evidence of shifts in the msg-I repertoires over time within the same patients while there are reports suggesting that this repertoire is stable in the same patients over a period of months. In addition, studies of *P. carinii* and *P. murina* have found a conservation of the msg-I repertoires over time and between strains. Direct proof of the hypothesis of new msg-I gene generation would require sequencing the complete msg-I repertoires using sequential samples collected at different time points from the same patients (ideally infected with a single strain).

9. While informative, the inclusion of data on the conservation of the upstream and downstream sequences for the non-Msg-1 gene families does not seem to fit in with the focus on the repertoire and expression of Msg-1 genes. No discussion of the complete repertoire or expression data for these other Msg genes among different isolates is provided. Such information would fit in more closely with the theme and focus of the paper. Do these other families show the same type of variation as seen in the Msg-1 family? Is there any evidence suggesting that they are all expressed or only a subset per organism at a given time? If a subset, is there a suggested mechanism for control of their expression,

given that each has its own leader and presumably promoter?

Minor comments:

1. The authors used a mixture of 2 Msg genes in a PCR reaction to test for PCR-related chimerism artifacts. Given that there are 80 or more Msgs per sample, it would have been helpful to run a PCR with more plasmid-derived Msg variants, to see if that impacted the level of chimerism artifacts. Similarly, the authors chose only 3 out of more than 900-1000 alleles identified by PacBio for validation by PCR. It would be helpful to confirm more alleles by PCR, particularly those with predicted mosaic sequences and those identified from single patients.
2. Southern blots could help address some of the postulated mechanisms for generating diversity. While this is likely difficult to do with *P. jirovecii* due to the low amounts of organism DNA in clinical samples, this could be done with other species, such as *P. carinii* or *P. murina*. Of note, duplication of subtelomeric regions has previously been identified in *P. carinii* and *P. murina*, supporting the proposed mechanism of rearrangement of the subtelomeres through single recombinations.
3. It would be helpful to provide the gene IDs/GenBank numbers for the site-specific recombinase genes used to search of the *Pneumocystis* genome, and explain how the absence of this gene relates to the role of CRJE in recombination.
4. The use of the term "complete repertoire" in many places is confusing given that only a subset of the complete repertoires was identified from the majority of samples. It may be better to use a different term to accurately reflect the identified alleles.
5. The msg-I dataset generated in this study is potentially useful to the *Pneumocystis* research community for further investigation of msg diversity and evolution in different patient populations. Thus, it would be helpful to provide all msg-I alleles identified from each patient, especially for patients with a large number of msg-I alleles identified. If it's difficult to provide them as a supplemental file, they could be deposited into an online database.

Reviewer #3 (Remarks to the Author):

This manuscript provides a large and impressive analysis of MSG gene diversity encoded by the important and genetically intractable fungus *Pneumocystis jirovecii*. The authors concentrate primarily on MSG family I, which has been proposed as being the most likely gene family involved in immune evasion by antigenic variation, given evidence that only one of an estimated 80 genes are adjacent to a promoter and, hence, likely to be transcribed at one time. Using PCR to recover expressed and silent copies of MSG 1 genes from 24 patients infected with *P. jirovecii*, the authors perform extensive analysis that reveals remarkable sequence variation, suggesting profound levels of change and reassortment. As such, the work provides very valuable insight into the MSG gene repertoire in this fungus and will act as an important reference dataset for understanding *Pneumocystis* immune evasion. I have a number of main concerns, and some smaller issues for the authors' consideration.

Main issues.

1. As stated above, this manuscript provides highly valuable information on MSG gene conservation and variation in static, geographically distributed samples. It cannot therefore directly inform on one aspect of immune evasion: MSG gene change over time during an infection, due to evasion of host adaptive immunity. It would be valuable for the authors to make this limitation clear in the discussion.
2. The authors correctly state (eg line 317) that the PCR approach used may result in artefactual gene chimeras. They state that "[any] errors were specifically addressed and are not believed to affect the

results presented below (see methods)'. However, the details of the tests run and provided in the methods is very limited and should be expanded upon. What two genes were tested; how similar were they; did the trimming resolve the issue with chimeras and, if so, how do we know it did? In addition to this, they state that only 5% of genes were pseudogenes (as defined by having stop codons); might the trimming have excluded truncated pseudogenes?

Several aspects of the paper, and in particular the discussion, are very speculative and overlong, and a number of elements should be reduced and potentially discarded:

3. The authors suggest that MSG gene variation arises during meiosis; indeed, this is one of the main conclusions in the abstract. However, I cannot see any evidence to suggest MSG recombination is limited to, or even mainly arises during, meiosis, and so it appears to be merely a suggestion. I would suggest this focus should be reduced considerably.

4. In the methods and the main text, the authors describe how they searched for recombinases, and refer the reader to supplementary data. No recombinases were found, and no detail is provided for what genes/proteins were looked for. As this is negative, it does not seem worth reporting.

5. The description of potential triplex forming sequences (Fig. 7) is very interesting. However, there appears to be little/no evidence that these actually contribute to MSG recombination, or that they form the proposed secondary structures. As such, these data should come with a caution and be limited in their presentation. As a small point, the authors suggest the triplex may not be conserved in other *Pneumocystis* species, including *P. carinii*; I am not an expert, is this not the old name for *P. jirovecii*?

6. 'putting in perspectives'. This section could be removed, as it summarises what has been said before.

Small issues:

Line 97. This can be removed, as it is not clear what relevance the PhD thesis has to the paper.
Figure 2. I'm afraid I do not follow this graph, and cannot connect it to the previous data: what is denoted by 'number of strains (1-5)', and how does this relate to the patients and/or sample locations?

RESPONSES TO THE REVIEWER COMMENTS

(NCOMMS-23-15687-T, Meier et al.)

Reviewer #1 (Remarks to the Author):

The manuscript describes the repertoires of *Pneumocystis jirovecii* major surface glycoproteins sequenced after PCR amplification from clinical samples. The findings essentially corroborate previous observations of *msg* expression and lead the authors to make some interesting hypotheses regarding molecular mechanism of antigenic variation in this fungus. I think, in terms of conclusions that may be sustained by the data, the findings represent only a modest advance (i.e. principally, how repertoires from different clinical samples compare) on what was already known (i.e. that there are distinct *msg* subfamilies with specific and fluid genomic dispositions, monoallelic expression, mosaic sequences etc). The comparison of clinical samples seems not to provide explanation for repertoire (in terms of sample provenance), but is more consistent across samples than would be the case for other antigenic variation systems. The novelty in the manuscript comes from the new ideas around molecular mechanism (i.e. subtelomeric reorganisation, the involvement of DNA triplexes). While plausible and attractive, these are not tested, and seem to derive from published genomic data not derived from the clinical samples. Overall, I think the work could go much further to address its own ideas, even without an in vitro culture system. Relying on sequencing of PCR products seems to me to be unnecessarily limited, and lacking internal corroboration by other methods. I suggest some extensions below that I do not think are beyond the scope of the questions posed, and which would represent significant advances of what we already know.

Major points:

1. Why is there no RNAseq of *msg* mRNA?

Response:

We previously reported RNAseq data of the six *msg* families for six patients with *Pneumocystis* pneumonia (Schmid et al 2021, reference 14 of the present manuscript, cited at lines 79, 80 and 88 in the introduction). To clarify the issue, now we have added the relevant observations of this study (NB: in the revised manuscript, new text is underlined, deleted text in struck through):

At line 79:

“During pneumonia, families I and III represent respectively ca. 85 and 10% of the *msg* transcripts, whereas the other families only 1 % each ¹⁴.”

At line 88:

“Genes of families II to VI each have their own promoter and could be constitutively and simultaneously expressed ¹⁴. We observed that several alleles of each family, except V and VI, are expressed during a single infection. However, it is not known if all or only some genes of each of these families are expressed in single cells. This regulation might occur during the cell cycle or by silencing because of the proximity of the telomeres (the “telomere position effect”)
¹⁵”

As mentioned by this reviewer in his second major point here below, single-cell RNAseq would provide data that may help understanding further the expression mode of the different *msg* families.

Major point 1, continued: Are the authors assuming that ‘expressed’ genes are exclusively associated with the previously observed expression site? (p19). Can they guarantee this? It should at least be confirmed independently with transcript sequences.

Response:

We can guarantee this. Indeed, the strict association of *P. jirovecii* msg-I transcripts with the expression site UCS present at a single copy per genome was demonstrated (Kutty et al 2001, reference 20 of the manuscript). This mechanism ensures mutually exclusive expression and is identical to that observed in *P. carinii* infecting rats (reference 16) and *P. murina* infecting mice (reference 17). To clarify the issue, we have now modified the text and cited reference 20 concerning *P. jirovecii* at line 95, at the end of the sentence starting at line 92:

“On the other hand, ~~a single one gene~~ one gene out of the ca. 80 genes of family I is ~~believed to be~~ expressed at a time in a cell thanks to the presence of a single copy promoter in the genome, within the so called upstream conserved sequence (UCS) ^{16,20}.”

2. Ideally, this would be single-cell RNAseq, providing much richer data on cellular heterogeneity in msg expression and directly quantifying the subpopulations the authors predict to exist. I see no reason why this should not be possible to develop from the starting material

Response:

Single-cell RNAseq has not been reported so far for *P. jirovecii* infecting humans, nor for *P. carinii* infecting rats or *P. murina* infecting mice although this would be easier given that more material is obtained from the animal model than from clinical specimens of patients. In fact, the first single-cell analysis of *P. carinii* will be reported at the 16th International Workshop on Opportunistic Protists (IWOP) to be held here in Lausanne in August 2023. Thus, although probably possible, as this reviewer points out, we did not embark on the development of such a demanding analysis on clinical specimens of patients, this would constitute a whole study.

3. Why is there no genome sequencing to provide new data on the dynamism of subtelomeric loci? Comparison of clinical samples would then confirm the subtelomeric recombination events that the authors imagine, as well as define the recombination breakpoints (p26), perhaps giving a novel mechanistic insight. This is possible from clinical samples, using cell sorting and single-cell sequencing. There are several commercial products to enrich sequences for known targets, such as *Pneumocystis* msg loci.

Response:

We agree that the approach suggested by this reviewer could be feasible. However, we did not embark on such a study because it would be complicated by two important parameters:

- (i) Co-infection with several *P. jirovecii* strains is present in most of the patients with *Pneumocystis* pneumonia. This prevents the differentiation and assembly of the subtelomeres of each strain separately. This is the reason why we previously analysed a patient infected with a vastly dominant single strain to sequence and assemble *P. jirovecii* subtelomeres (Schmid et al 2017, reference 7). Consequently, sequencing of the subtelomeres was not planned in the project that led to the present manuscript. And, as expected, co-infections turned out to be present in 15 of the 24 patients analysed.
- (ii) Obtaining sequential samples from the same patients has become very difficult nowadays. The situation was different thirty years ago, before HAART has been implemented. In

these times, the number of PCP cases was high, and such sequential samples from the same patients were frequent.

The enrichment in known targets suggested by this reviewer is indeed a valuable tool. In our RNAseq analysis (Schmid et al 2021, reference 14), we did use enrichment in *msg* transcripts using oligonucleotide probes.

4. Why is there no formal analysis of recombination (i.e. some kind of phylogenetic profiling, GARD, TOPALI, PHI-Pack etc.) to distinguish mosaicism from alternative explanations such as independent sorting of divergent gene duplicates (time series RNAseq would also provide a critical perspective on this). I wasn't sure how the authors defined a mosaic here, but it seemed to involve BLAST only.

Response:

We previously used the TOPALI, Recombination Analysis Tool, and Bellophoron methods of detection of recombination events, and evidenced that recombinations occur strictly between genes of the same *msg* family (Schmid et al 2017, reference 7). The recombinations were detected by similar fragments shared by the mosaic gene and its putative parent genes. In about one third of the case, the shared fragments were identical and thus duplicated. Because the methods of recombination detection allow the analysis of only a limited number of genes, we used in the present manuscript the BLASTn algorithm to detect duplicated gene fragments within the large set of *msg*-I alleles we identified. To clarify the issue, we have now added at line 556 in the Results:

"We used the BLASTn algorithm because it allows the analysis of a large set of alleles."

The definition of gene mosaicism that we used in the present manuscript is given at line 552: "The mosaicism of *msg* genes, i.e. being composed of fragments potentially originating from other genes, was suggested based on the detection of recombinations and duplicated fragments strictly among members of the same family ^{7,18}."

This definition of the mosaicism does not encompass the mechanisms that created the mosaic genes. This reviewer suggests that the "independent sorting of divergent gene duplicates" might be such a mechanism. Another mechanism could be double intragenic recombinations leading to exchange or conversion of gene fragments, conversion leading to duplicated fragments. At line 761 of the discussion, we mention that our analysis reported in Figure 6 suggests a further mechanism, i.e. the duplicated fragments might have been conserved from ancestral alleles during the diversification of the alleles. We hypothesize at lines 777 to 782 that the ancestral alleles may be split by recombinations involved in fragment exchanges and/or rearrangements of subtelomeres.

5. Why not follow up the smaller group of *msg*-I that occupy expression-sites in all patients: are they surface expressed? Presumably you can raise an antibody to them, and obtain Pneumocystis from clinical samples by cell sorting?

Response:

Such experiment has been performed with *P. carinii* infecting rats and evidenced focal surface expression of three *msg* variants in a single host lung (Angus et al 1996, J Exp Med). This has not been reported for *P. jirovecii* infecting humans so far, probably at least in part because working with clinical human samples is more difficult than with samples from an animal model.

Nevertheless, such experiment might be feasible in *P. jirovecii*, it constitutes a whole study in which we did not embark.

6. Why is there no serology? I would imagine that host antibodies are present in some clinical samples? Testing the serological response to observed msg using simple methods would test the idea that rarer alleles are less immunogenic (such frequency dependency being critical to antigenic variation).

Response:

We fully agree that the suggested serology studies might be fruitful to better understand the antigenic variation system. It has not been reported for any *Pneumocystis* species so far. We think that this is valuable study that could be carried out in the future.

Minor points:

1. Please use line numbers

Response: This is now done throughout, including in Supplementary data file.

2. P2, l13: "The recombination also generates new mosaic genes constantly"

Response: This is now corrected at line 50.

3. P3, l15: remove "it"

Response: It is now deleted at line 77.

4. P3, l22-3: "recombination of genes in this genomic region is advantageous as it has no, or little, effect..."

Response: This has been now modified as suggested at lines 85-86.

5. P4, l1: replace "single" with "one gene"

Response: This is now done at line 93.

6. P4, l5 "...is also found at the start of each msg-I allele"

Response: "existing" is now replaced by "found" at line 97.

7. P9, l6: should this sentence read "so far DOES NOT account for...?"

Response: no, to clarify the issue, "accounts for" has been replaced by "encompasses" at line 235.

8. P9, l15: "...sequencing technique used."

Response: This is now modified at line 245.

9. P9, l17: "...present IN patients..."

Response: "in" has been now added within the tile at line 248.

10. P9, l22: "...amplify each of them specifically (Table S3)."

Response: This is now modified at line 253.

11. P11, l12: "BLASTn analyses"

Response: Thank you, this is now corrected in the title at line 298.

12. P12, l4: "Duplicated fragments were used to search..."

Response: Accordingly, we have modified the sentence at line 315 as follows:

“Duplicated fragments ~~were searched~~ were used to search mosaicism within the...”.

13. P12, paragraph 2: I am unsure from the text what exactly the purpose of this activity is. What exactly were you looking for in your sequences. How did you define what was a mosaic?

Response:

We have answered to this point in our response to the major point 4 of this same reviewer, here above.

14. P15, l11: what is meant by “related ones”? A better expression would be “There were no clades of *msg-I* sequences in our trees that could be associated with a specific repertoire, suggesting that sample provenance does not explain the tree topology.”

Response:

We agree and have replaced the sentence by one that is very close to that suggested by this reviewer at line 377:

“~~No repertoire showed clear groups of alleles that would have revealed the presence of related ones.~~ There were no clades of *msg-I* sequences associated with specific repertoires, suggesting that sample provenance did not explain the tree topology”.

15. P16, l8: ‘Conversely’, not “reversely”.

Response:

This is corrected now at line 422.

16. P18, l5: ‘Consistent’ not “consistently”

Response:

This is corrected now at line 472.

17. P18, paragraph 2: I think the use of homo/heterogeneous here is not quite correct. A situation where most abundance belongs to just a few alleles would seem to be more homogeneous than a situation where distribution is more equitable, but you seem to be suggesting the opposite. In any case, I think you mean that abundance has an overdispersed distribution (most sequences come from a few types) in expressed repertoires, and a more normal/underdispersed distribution in complete repertoires.

Response:

We agree, this is now corrected as suggested by this reviewer at lines 490 and 492: “underdispersed” and “overdispersed”, respectively.

Accordingly, “heterogeneity” has been replaced at line 503 as follows:

“...~~heterogeneity~~ overdispersion...”

18. P22, l3: “To understand the phenomenon better,”

Response:

Thank you, this is now corrected as suggested at lines 574.

19. P22, l10: “The 33bp CRJE sequence...”

Response:

This is now corrected as suggested at lines 582.

20. P32: Plasmodium

Response: Thank you, this is now corrected at line 822.

Reviewer #2 (Remarks to the Author):

The authors have evaluated the diversity of the Msg-I gene family repertoire in *Pneumocystis jirovecii* and explored the level of conservation among the other Msg gene families. They also describe hypothetical mechanisms for recombination in the Msg-I gene family. This is the most extensive and detailed analysis of the Msg-1 repertoire among patient isolates conducted to date, and helps clarify the broad diversity of the repertoire, the geographic distribution of individual alleles, and the importance of recombination in generating Msg-I diversity.

While the study provides important insights into the organization of the Msg genes, there are a number of issues to consider that can potentially improve the manuscript.

Major comments:

1. A major focus of the paper is to characterize the Msg-I repertoires of *P. jirovecii* strains. However, this analysis is greatly complicated by the fact that the majority of patients were infected by multiple strains, making it difficult to describe the repertoires in single strain/organism. Particularly, with the methods presented, it is impossible to determine if the msg sequences identified in these patients (with multi-strain coinfection) were from the same or different strains. This caveat, coupled with the fact only a subset of the Msg-I repertoires were obtained from the majority of patients, limits the depth of insights into the population structure, diversity and potential recombination patterns of the msg-I genes.

Response:

This is indeed a difficulty that cannot be avoided. Its impact on the results and their analysis is discussed in our responses here below.

2. The authors suggest that all the Msg genes in the *Pneumocystis* genome are being amplified by the genomic PCR (the complete repertoire), but their own data suggests that this is not the case for all samples. If the entire repertoire is being captured in each PCR reaction, replicate amplifications should be identical, but as shown in Table S5, this is not the case for either the complete or the expressed repertoire. While low abundance may explain the discrepancy in the expressed repertoire, this should not be the case for the entire repertoire, especially those samples with a single strain, as each Msg gene should be present in equal numbers, as a single copy per genome, with an expected ~80 copies/genome. In samples with more than one strain, there should be an equal abundance of the ~80 Msg genes for each strain, although the strains may be present in varying proportions. Assuming that each organism has a similar number of Msg genes per genome, it is surprising that many samples have substantially fewer alleles identified.

Response:

To answer to this highly relevant and crucial comment, we discuss some issues in the following paragraphs.

Our genotyping using PacBio CCS sequencing of the ITS1-5.8-ITS2 region suggested the presence of a single strain in nine of the 24 patients analysed in the study (Table S5). To further determine the number of strains in these nine patients, we have now applied the conventional genotyping method consisting in subcloning PCR products of four markers followed by Sanger sequencing of subclones to detect co-infections. The results agreed with the presence of a single strain in three patients (LA2, CI5, BE2), whereas the other six harboured at least two alleles of at least one marker, suggesting the presence of at least two

strains (new Supplementary data 2 and new Table S7). Note that, consistently, the ITS1 alleles observed with the two genotyping methods were identical, except only for patient LA9 that harboured in addition the allele B2.

As mentioned by this reviewer, the three patients infected by a single strain should have the *msg-I* alleles composing the complete repertoire in equal abundance in their DNA sample, *i.e.*, before the PCR reaction. Indeed, previous studies have shown that most of these alleles, generally all, are present each at a single copy per genome (we observed only one duplicated *msg-I* allele among the subtelomeres of a single strain, see Supplementary data 4; and none were observed in *P. jirovecii* by Ma et al 2016, reference 4). The abundances of each allele after the PCR and PacBio sequencing observed in the patients LA2, CI5, and BE2 infected by a single strain ranged from 0.4 to 6.1% of all reads composing the given repertoire (Supplementary file *msg-I_alleles_abundance_in_patients.xlsx*). This variation might result from the exponential process of PCR that increases the stochastic differences between the number of alleles used as templates at the beginning of the reaction. Nevertheless, this latter phenomenon is unlikely to be an important parameter here because we observed in our reproducibility experiment that the varying allele abundances of the complete repertoires were fairly reproducible in duplicate analyses (Figure S4). The co-infecting strains present in different abundances in most patients can account for such reproducible varying abundances of the alleles. However, this is not the case for sample LA2 because it most likely harbours a single strain, as revealed by the second genotyping method. This latter observation could be explained by the two following hypotheses:

- (i) The different alleles are not amplified and/or sequenced by PacBio with equal efficiency, rendering some alleles difficult to detect. This further suggests that an unknown proportion of alleles might even not be detectable at all. We did not detect any notable differences between the alleles of LA2 that could support this hypothesis, in size nor using YASS software (<https://bioinfo.univ-lille.fr/yass>) or other tools to detect motifs (see <https://molbiol-tools.ca/Motifs.htm>). This first hypothesis could explain that the three patients CI5, LA2, BE2 harboured respectively only 44, 49, and 56 alleles, which is below the value of 80 for a single strain reported in the literature. It could also explain that four of the other six patients that turned out to harbour two strains rather than one harbour only 54, 59, 61, and 79 alleles (respectively LA7, LA9, SE1, and LA3). The two remaining samples LA8 and SE2 had more than 80 alleles, *i.e.*, 105 and 115, which also appears too low for two strains.
- (ii) The reproducible lowest abundances are due to a supplementary co-infecting strain present at a low abundance that remained undetected because of the insufficient discrimination power of the genotyping methods. Moreover, the co-infecting strains should share many *msg-I* alleles to explain the low number of alleles in presence of two strains. This has never been reported so far but cannot be excluded. Contrary to the first, this second hypothesis could not explain that the three patients infected by a single strain and four by two strains harboured less than 80 alleles.

Although the variation of the number 80 of *msg-I* genes per genome is presently unknown, the first hypothesis appears more likely than the second one because it can better explain the results. Thus, it is possible, although not proven, that the numbers of alleles observed in the complete repertoires that we report are underestimated to an unknown extent, as suggested by this reviewer. Importantly, this underestimation would not impact any of the conclusions drawn in the study because none is based on the absolute values of these numbers.

In our reproducibility experiment, the low abundant alleles account for 100% of the differences between duplicate analyses of the complete repertoires. It turns out that these alleles might be those that are detected with a poor efficiency, and that other alleles might not be detectable at all by our methodology. This hypothesis has been now integrated in the text (see below the modifications made).

According to these considerations, we have now clarified the issue by adding the following sentences and making the following modifications:

- These results of the second genotyping method used are now described in the new Supplementary data 2 and Table S7 as follows:
“6. Multitarget genotyping to assess infection by a single *P. jirovecii* strain
We further investigated the number of *P. jirovecii* strains present in the nine patients infected by a single strain according to the PacBio ITS1-5.8-ITS2 genotyping. We used the genotyping method consisting in subcloning PCR products of four markers followed by Sanger sequencing of subclones to detect co-infections^{18,19}. The presence of two alleles of at least one marker for six of the nine patients revealed the presence of at least one supplementary co-infecting strain (Table S7).”

“Table S7. Alleles detected using multitarget genotyping of *P. jirovecii* ^a”

Consequently, references 18 and 19 have been added in the list for Supplementary data:

18. Hauser, P.M., Francioli, P., Bille, J., Telenti, A., Blanc, D.S. Typing of *Pneumocystis carinii* f. sp. *hominis* by single-strand conformation polymorphism of four genomic regions. *J. Clin. Microbiol.* 35, 3086-3091 (1997).
19. Gianella, S., Haeberli, L., Joos, B., Ledergerber, B., Wüthrich, R.P., Weber, R., Kuster, H., Hauser, P.M., Fehr, T. & Mueller, N.J. Molecular evidence of interhuman transmission in an outbreak of *Pneumocystis jirovecii* pneumonia among renal transplant recipients. *Transpl. Infect. Dis.* 12, 1-10 (2010).

- At line 393:
“A reproducibility experiment showed that the varying abundances of the alleles of the complete repertoire harbored by patient LA2 infected by a single strain were reproducible in duplicate analyses (Supplementary data 1, Table S6, Figure S4). This was unexpected because these alleles are all present at a single copy per genome and thus in equal abundance in the DNA sample. This might result from (i) a varying efficiency of amplification and/or PacBio sequencing of the alleles, some being not amplifiable at all, and/or (ii) an undetected co-infecting strain present in low abundance. The first hypothesis is more likely because an underestimation of the numbers of alleles would explain better the results. Indeed, it would explain that the patients assessed to be infected by a single strain using a second genotyping method harbored in their complete repertoire less than the 80 alleles reported in the literature (44, 49, 56 in respectively CI5, LA2, BE2; Supplementary data 2, Table S7). It would also explain that four of the other six patients that turned out to harbour two strains rather than one harbour only 54, 59, 61, and 79 alleles (respectively LA7, LA9, SE1, and LA3). Importantly, this underestimation would not impact the conclusions drawn because none is based on the absolute values of these numbers. Figure 2 also shows the consistency of the data. Indeed, the correlation was 0.74 between the number of alleles of the complete repertoires and that of strains. Moreover, the average of 77.6 (intercept + regression

~~slope) alleles in samples infected by a single strain obtained by regression is compatible with an underestimation of the postulated number of 80 per genome."~~

- At line 419:
 "...probably because of the limitations of the methodology (see above) that affect the composition in low abundant alleles of the repertoires, and that lead to a slight underestimation of the number of these alleles (Supplementary data 1, Table S6, Fig-S4)."
- Supplementary data 1, from line 35:
 "This might result from a varying efficiency of amplification and/or PacBio sequencing of the alleles, some being not amplifiable at all (see Results). Thus, the number of alleles in each repertoire might be underestimated to an unknown extent. This can be explained by the stochastic variation in the number of copies of the low abundant genes used as template in the PCR amplification. In other words, low abundant alleles are not amplified each time in repeated PCRs."
- To clarify the results, we have now identified the three patients assessed to be infected by a single strain thanks to black stars on Figures 1a, 1b, and 6. Accordingly, we have added the following sentence in the legend of these two Figures: "The black stars identify the three patients assessed to be infected by a single strain (Supplementary data 2)."
- Figure S4 has been clarified by identifying better the duplicates on the left and separating their results with spaces. Its legend was completed as follows:
 "The bottom duplicate of each sample is also shown in Figures 1a and 1b. Patient LA2 was infected by a single strain."

Major comment 2, continued: Moreover, as the authors note, all the expressed Msgs should be identified in the complete repertoire, but again this is not the case. The authors again suggest that this is due to low abundant alleles, but as above, this does not seem to explain the incompleteness of the genomic repertoire.

Response:

In our reproducibility experiment, the expressed repertoires were less reproducible than the complete ones. This probably results from the fact that the PCR reaction starts with a very varying number of template molecules, corresponding to the varying sizes of subpopulations expressing each a specific allele. The exponential process of PCR might increase these variations, as well as increase the varying efficiency of amplification and/or PacBio sequencing of the alleles. Accordingly, we have complemented the Supplementary data 1 as follows at line 52:

"Moreover, the varying efficiency of amplification and/or PacBio sequencing of some alleles may also increase the phenomenon."

Major comment 2, continued: PCR for both the complete and the expressed repertoires relies on a previously reported primer which is located in the 3' UTR of the Msg-I genes. While the primer sequences appear to be well conserved, do the authors have evidence supporting that this 3' UTR region is linked to all Msg-I genes and is conserved among different *P. jirovecii* strains, i.e. that it can in fact be used to amplify the entire repertoire among all isolates? It would be helpful to note that an alignment of this primer with all msg genes containing the 3' UTR (including the primer binding region) that were not PCR generated verified this.

Response:

The degenerate site (wobble) within the reverse primer used and that is mentioned at line 143 of the methods is based on the alignment of all the 48 published *msg-I* sequences that include the region downstream the stop codon covering the primer (new Figure S1). This primer is identical to that previously used in reference 18, including the wobble. Only one of the 48 sequences might be less efficiently amplified because of the presence of a G instead of a A at the 3' end of the primer.

To clarify the issue, we have now completed the text and added this alignment as Figure S1 as follows at line 140:

“(GK452: 5' AATGCACTTTCMATTGATGCT 3'; the underlined M was introduced because ca. 90 and 10% of the sequences harbor respectively T and G at this position; this primer is identical to that previously published¹⁸, including the wobble). The alignment of the 48 published sequences including the region is shown in Fig. S1, only one might be less efficiently amplified because of the presence of a G instead of a A at the 3' end of the primer.”

Consequently, Figures S1 to S11 have been renumbered throughout the manuscript.

Major comment 2, continued: Whether these primers are able to amplify only partial or all *Msg* genes could also be examined by Illumina NGS of total genomic DNA (rather than a PCR product) to look for polymorphisms in the regions of the primers. Even if the primers are not fully conserved, the study still provides important insights into the subset of *Msg-1* genes that can be amplified.

Response:

The primer within the CRJE and the reverse one are fully conserved, except possibly the latter for ca. 2% of the genes (see just above our response to the third part of comment 2).

Consequently, the generic PCRs should amplify most *msg-I* genes present in an infection.

Nevertheless, as discussed here above in our response to the first part of comment 2, some alleles might be poorly or not at all detected by the methodology because of a less efficient amplification and/or sequencing by PacBio. This might have led to an underestimation of the number of alleles in each repertoire to an unknown extent, although this is not proven.

Major comment 2, continued: Furthermore, the efficiency of PCR to amplify the *Msg-I* repertoires could be greatly affected by the *P. jirovecii* DNA concentrations in clinical samples. To avoid or reduce the possibility of a partial or biased amplification, it would be helpful to determine the minimal *Pneumocystis* DNA concentration (ideally by qPCR to quantify the *P. jirovecii* genome or *msg* copy number) in clinical samples that would allow an amplification of the entire *msg-I* repertoires.

Response:

We have now added the concentrations observed in 22 of the 24 patients by realtime PCR specific to *P. jirovecii* within Table S1. The correlation between the fungal loads and the mean numbers of alleles in the complete repertoires was negligible (Pearson correlation coefficient 0.15, negligible is from -0.30 to 0.30 [Hinkle et al. Applied Statistics for the Behavioral Sciences. 5th ed. Boston: Houghton Mifflin, 2003], p-value = 0.50, n=22). That for the expressed repertoires was low positive but not significant (0.39, low positive is 0.30 to 0.50, p-value = 0.08).

To clarify the issue, we have now added the following sentences at line 382:

“The number of alleles of the complete and expressed repertoires were not significantly associated with the fungal loads present in the samples (Pearson correlation respectively 0.15 and 0.39, p-value = 0.50 and 0.08, n=22; data in Tables S1 and S5).”

The determination of the minimal concentration avoiding bias would be difficult. Indeed, several PCR products of samples with a fungal load below $1.0E+07$ *P. jirovecii* genomes per ml had to be pooled to obtain enough DNA for PacBio CCS sequencing (up to 20 PCR products for the complete repertoire of BR1). This also results from the fact that we used a small number of PCR cycles to minimize artefacts. Thus, it would be very tedious, to not say impossible, to dilute samples to determine the fungal load providing unbiased complete repertoires. In future experiments, random DNA amplification before performing the PCRs could be investigated to (i) obtain sufficient DNA amounts without pooling several PCR products and (ii) alleviate the need of a minimal fungal load for obtaining an unbiased complete repertoire.

To clarify the issue, we have now added the following sentences at line 174:

“Several PCR products from some samples with less than $1.0E+07$ *P. jirovecii* genomes per ml had to be pooled after the E-gel step to obtain enough DNA for PacBio sequencing.”

3. The authors have trimmed ~1,000 bp (~500 bp at either end) prior to conducting their analysis, to minimize inclusion of artificial chimeric Msgs in their analysis. However, the trimming was not made at the same positions for all Msg sequences, presumably due to the presence of indels. For example, for the Msg sequences from patients LA1, LA3 and BE1, the 5' ends of the retained sequences had 33-55bp size differences between the shortest and longest ones while the 3' ends of the retained sequences had 148-165bp size differences between the shortest and longest ones,. It would be beneficial to clarify the reasoning behind the uneven trimming and explain how this was factored into downstream analysis. A drawback of this trimming approach (1/3 of the full length) is that it may result in a further reduction in the number of unique alleles; alleles that appear identical may in fact not be identical since there can be substantial variation in the trimmed sequences, especially in the first ~500 bp. This should be taken into account when they draw their conclusions.

Response:

Because the chimeras observed upon mixing two *msg-I* alleles in a single PCR had all their extremities in the first or last 500 bps of their ca. 3.1 kb sequence, we trimmed all *msg-I* sequences obtained exactly from their position 500 to their position 2500 (this is explained at line 201 of methods). The indels present in their sequences, which generates the length differences of the trimmed sequences mentioned by the reviewer (these indels do not interrupt the open reading frame of the *msg-I* genes and contribute to the allele variation). The differences in length of the aligned parts of the sequences do not affect the downstream analyses because they are negligible relatively to the 1.8 to 2 kb that are aligned. They affect marginally the trees of classification of the alleles shown in the Figures of the manuscript, and not the conclusions drawn.

The drawback of missing the variation between the alleles that is present in the first and last 500 bps of the *msg-I* sequences cannot be avoided. Indeed, we do not know the proportion of the artefactual chimeras in our sequences, rendering the trimming necessary. When the full ca. 3.1 kb sequences are analyzed instead of the 2kb ones, 69% more alleles are identified (1702 versus 1007). This increase results in a decrease of the overlaps of the complete and expressed repertoires (respectively 67 versus 84% and 54 versus 61%, see lines 436 to 439 of the results) and to an increase of the proportion of the alleles of these repertoires observed in a single city (respectively 68 versus 51% and 88 versus 73%, see Figure 4). The latter alleles are mostly in single patients in slightly increased proportions (respectively 90 versus 88 and 93

versus 89%, see Figure 4). The proportions obtained with the 3.1 kb sequences would not change the conclusions drawn in the manuscript, *i.e.*, that the alleles are shared between patients from different locations and that a high proportion of the alleles identified are present in single cities and patients.

To clarify the issue, we have now added the following sentences at line 207:

“This trimming reduced the diversity of the alleles identified by ignoring some variation but was necessary to avoid the artefactual chimeras. It affected marginally the results presented and had no impact on the conclusions drawn.”

In addition, we corrected the related following errors at line 439:

“Indeed, $84\% \pm \text{SD } 7\%$ of the alleles of each complete repertoire were present in at least one complete ~~or expressed~~ repertoire of another patient. The value was $61.77\% \pm \text{SD } 19.16\%$ for the expressed repertoires.”

4. On page 23 the authors speculate that the RA in the CRJE peptide may serve as a target for host peptidases to remove the constant part of Msg proteins to maximize antigenic variation. The authors, however, provide no data to support this hypothesis, and it seems questionable for a couple of reasons. First, the RA site is not conserved across *Pneumocystis* species. Given that the *Msg-I* gene family system has been conserved across all *Pneumocystis* species examined to date, and thus presumably dates back to a primordial ancestor of all *Pneumocystis* species, it's likely (though not certain) that the mechanisms for ensuring antigenic variation are similar across species. Moreover, if host proteases are processing, then host immune cells would have access to the fragments as well and could potentially develop an immune response to the fragments, potentially to the detriment of the organism. Experimental data need to be provided to support this hypothesis.

Response:

In our opinion, the antigenic variation systems of the different *Pneumocystis* species may have evolved differently from the common ancestor. Indeed, this might result from the varying selective pressures by the immune systems of the different mammal host species. The hosts are quite different. In fact, although they are similar, it is already clear that these antigenic variation systems do present differences, for example in the structure of the CRJE conserved sequences present at the beginning of each *msg-I* genes (see below). Consequently, we have added the following sentence at line 618:

“This may have evolved specifically within the human host because the CRJE sequences of *P. carinii* and *P. murina* do not encode such motif.”

Our hypothesis is that the RA motif might be recognized by some peptidase. We agree that recognition by a host peptidase is less likely, as suggested by this reviewer. Consequently, we have deleted this speculation at line 614 of the results:

~~“, for example the transmembrane serine protease present in the human lungs that is involved in the defence system (Uniprot O60235).”~~

In addition, we corrected the related following errors:

At line 592:

~~“...so-called *H-DNA, so-called *H-DNA (Fig. 8b, the symbol * stands for Hoogsteen bonds).”~~

And at line 599:

~~“...frequent reported to constitute *H-DNA (CG*G, the symbol * in the triad stands for Hoogsteen bonds).”~~

5. Similarly, on page 22 the authors hypothesize that the CRJE region contains DNA elements that allow formation of a DNA triplex, and suggest that this triplex may play a role in translocation of entire *Msg* genes. They note however, that the CRJE of *P. murina* is unlikely to form such a triplex, and that of *P. carinii* could possibly form one, though it has a less symmetrical mirror repeat. Again, given that the *Msg* system for presumed antigenic variation has been maintained across *Pneumocystis* species since its origin in a common ancestor, it seems likely that a mechanism for recombination would also have been maintained. It would again be important to have experimental support of this hypothesis to include it in the manuscript. The current conclusion in the discussion that “Reassortment of the *msg-I* genes’ repertoires and exchange of the expressed allele by translocation of entire genes mediated by DNA triplexes” is not warranted specifically as related to DNA triplexes.

Response:

As for the RA motif evoked just here above, we do think that it cannot be excluded that the formation of DNA triplexes to mediate the recombinations has evolved from the common ancestor only within *Pneumocystis* species infecting humans and rats. Alternatively, it could have been lost in *P. murina* infecting mice. Indeed, the selective pressures in these hosts might be different.

Although we miss experimental data to support the triplexes hypothesis, the presence of ca. 80 copies per genome of the mirror repeat within the CRJE sequences constitutes *per se* a strong evidence of the importance of this sequence. Indeed, (i) it cannot be so numerous by chance, (ii) it is localized exactly at the position where recombination has been postulated to occur, and (iii) triplexes are known to mediate recombination. Drawing the hypothesis that triplexes are playing a role in recombinations involved in the antigenic variation system of *P. jirovecii* may foster research on this topic. The investigation of these potential DNA triplexes is a whole study.

We fully agree that the sentence mentioned by this reviewer concerning the triplexes is too conclusive in the discussion. Accordingly, we have now deleted twice “mediated by DNA triplexes” at lines 635 and 640:

“(i) Reassortment of the *msg-I* genes’ repertoires and exchange of the expressed allele by translocation of entire genes ~~mediated by DNA triplexes.~~”

6. There are inconsistencies in the number of alleles identified in the study. Page 12 noted that 917 alleles were identified in the complete repertoires, while the last line on page 9 notes that 1007 alleles were identified. The authors note on page 15 and in Table S4 that the complete repertoire contained 44 to 185 alleles, and the expressed repertoires contained 2 to 108 alleles, but later note, line 388 and Table 1, that one sample had 411 unique alleles in the complete repertoire, and similarly that the same sample had 347 expressed alleles. However, none of the other tables or figures (e.g. Figure 1, Table S4) include a sample with that many alleles. Figure 4A and 4B similarly noted that e.g. 88% and 88.5% of the alleles in single cities were present in a single patient sample, but the graphs do not seem to identify a patient with that predominance of alleles. This inconsistency needs to be clarified. For the sample with 411 unique alleles, how many *Pneumocystis* strains were identified in the sample?

Response:

The number of distinct alleles identified in our study is indeed 1007, *i.e.*, 917 in the complete repertoires plus 90 identified only in the expressed repertoires. To clarify the issue, we have now completed the text at line 254 as follows:

“...the 1007 different *msg-I* alleles identified in the complete plus expressed repertoires observed in the present work,...”

No patient harbored 411 and 347 unique alleles, as mentioned by this reviewer. In fact, 411 and 317 alleles were identified in single patients, but different single patients.

To clarify the issue, we have now added the following text at line 450:

“Thus, each of the 24 patients harbored several of the 411 alleles that were observed in only one patient.”

Major comment 6, continued: Table S4 also lists two patients infected with a single strain (LA8 and SE2); each of them had 105 or 115 alleles, which are substantially more than other samples and the expected total number of ~80 per strain. Although it is likely that the total number of alleles varies among different strains, given that these are substantially more than anticipated in a single strain, it would be important to rule out the possibility of inappropriate classification of those alleles. Are the sequence divergence levels among those alleles in each sample significantly lower than those in other samples or are there any unique sequence patterns in those samples? Another possibility is that these represent coinfection with multiple strains which were indistinguishable by the genotyping method used. It may be worth genotyping using additional markers with high discriminatory power. It would be helpful to add GenBank numbers to Table S4 for all genotypes identified.

Response:

As explained above in our response to the first part of the major point 2 of this reviewer, here above, we have now used a second multitarget genotyping method. It revealed that six of the nine patients infected by a single strain according to the first genotyping were in fact infected by at least one supplementary strain. This might explain that patients LA8 and SE2 harbored more than 80 alleles in their complete repertoire.

The sequence divergence among the alleles that are present in the three patients assessed to be infected by a single strain is not different from that observed in co-infected patients. Indeed, they are uniformly spread along the classification trees of the alleles (Figure 1a, the three patients assessed to be infected with a single strain are now identified in this Figure by black stars).

As suggested by this reviewer, we have now added in a new column of Table S5 the ITS1-5.8S-ITS2 genotypes observed in the patients. In addition, their known GenBank numbers are given in the new Table S4 and the sequences of the 25 genotypes observed are in the new supplementary file ITS1-5.8S-ITS2_alleles.fasta.

Accordingly, we have added the following sentences at line 294:

“The sequences of the 25 genotypes observed are in the Supplementary file ITS1-5.8S-ITS2_alleles.fasta and their correspondence to those present in GenBank in Table S4.”

Consequently, Table S4 to S10 have been renumbered throughout the manuscript.

Besides, to clarify the results of the genotyping, we have added the following sentence at line 268:

“For the sake of clarity, we use in the present manuscript the term “strain” despite that the correspondence of each genotype identified to a biological strain cannot be ascertained in absence of culture *in vitro*.”

7. It would be helpful to know the organism load in each sample, based on qPCR, if that is available. Normalizing the *P. jirovecii* concentration in all samples or at least those with high organism burdens may help reduce the variation in the number of alleles though this may be challenging due to the presence of coinfection with multiple strains. Did the samples with a low expressed repertoire (e.g. BR1, BR2, and BR3) have a low overall organism burden? If so, that might explain the low expressed repertoire. If not, is there anything unique about the patients from whom these samples were obtained? It’s noteworthy that they represent at-risk patients with 3 different underlying diseases (giant cell arteritis with presumed corticosteroid therapy, cancer, and HIV infection).

Response:

Samples BR1 and BR2 with respectively only three and two alleles in their expressed repertoire contained indeed a relatively low fungal load (respectively 1.98E+06 and 1.02E+06, Table S1). However, BR3 with only five alleles had a load of 6.98E+07 that is close to the average of 1.10E+08 for all 22 samples with known loads. Moreover, LA9, BR4, and SE3 with respectively only seven, nine, and eight alleles had load of 3.66E+06, 3.82E+06, and 1.15E+06, whereas LA7 with nine alleles had 6.10E+07. In fact, there was no significant association between the fungal load and the number of alleles of the expressed repertoires (described here above in our response to major comment 2 of this reviewer, Pearson correlation 0.39, low positive correlation: 0.30 to 0.50, p-value = 0.08). Thus, there was not anything unique to samples BR1, BR2, and BR3, except that they all come from Brest in France, which might suggest a possible geographical influence of the composition of expressed repertoire.

To clarify the issue, we have now completed the text as follows at line 387:

“..., but no correlation with the fungal load or underlying disease was observed.”

8. While it’s clear that mosaicism plays a role in generation of *Msg* diversity, the time-line for this is unknown; the conclusion that there is constant generation of new mosaic genes (in the Abstract) is hypothetical but currently not supported by experimental evidence. Despite the great variability of the *msg-I* repertoires among unrelated *P. jirovecii* isolates, there has been no evidence of shifts in the *msg-I* repertoires over time within the same patients while there are reports suggesting that this repertoire is stable in the same patients over a period of months. In addition, studies of *P. carinii* and *P. murina* have found a conservation of the *msg-I* repertoires over time and between strains. Direct proof of the hypothesis of new *msg-I* gene generation would require sequencing the complete *msg-I* repertoires using sequential samples collected at different time points from the same patients (ideally infected with a single strain).

Response:

We agree that the time-line of the *msg-I* genes mosaicism is presently unknown. Accordingly, we have modified the sentence at line 50 of the Abstract as follows:

“The recombinations also generates ~~also~~ ~~constantly~~ new mosaic genes.”

We also agree that sequential samples from the same patients will be required to investigate the speed of evolution of the *msg-I* repertoires in *P. jirovecii*. Accordingly, we have now completed our statement about the issue at line 811 in the section “Putting in perspective” as follows:

“However, but their frequency of these recombinations and the speed of the subtelomeres evolution over time remain to be determined, possibly by the analysis of sequential samples from the same patients.”

9. While informative, the inclusion of data on the conservation of the upstream and downstream sequences for the non-Msg-1 gene families does not seem to fit in with the focus on the repertoire and expression of Msg-1 genes. No discussion of the complete repertoire or expression data for these other Msg genes among different isolates is provided. Such information would fit in more closely with the theme and focus of the paper. Do these other families show the same type of variation as seen in the Msg-1 family? Is there any evidence suggesting that they are all expressed or only a subset per organism at a given time? If a subset, is there a suggested mechanism for control of their expression, given that each has its own leader and presumably promoter?

Response:

Because of the absence of small fully conserved regions up and downstream of the genes of the families other than family I, we could not investigate the diversity of alleles of these genes among patients by the PCR approach used for the family I. Nevertheless, our analysis of the up- and downstream regions of these families shows that these regions are also similar, which suggest the hypothesis that translocation of entire genes of these families might occur as for family I. This is stated at line 661 of the discussion. It is also stated there that no experimental data are presently available test the hypothesis of these translocations.

We previously used RNAseq to investigate expression of the genes of the families other than family I (reference 14). The findings are described in the introduction at lines 79 and 87 of the introduction. Following the questions raised by this reviewer about the issue, we have clarified the issue by completing our statement at line 88 as follows:

“We observed that several alleles of each family, except V and VI, are expressed during a single infection. However, it is not known if all or only some genes of each of these families are expressed in single cells, possibly by a regulation during the cell cycle or by silencing because of the proximity of the telomeres (the “telomere position effect”) ¹⁵.”

Minor comments:

1. The authors used a mixture of 2 Msg genes in a PCR reaction to test for PCR-related chimerism artifacts. Given that there are 80 or more Msgs per sample, it would have been helpful to run a PCR with more plasmid-derived Msg variants, to see if that impacted the level of chimerism artifacts. Similarly, the authors chose only 3 out of more than 900-1000 alleles identified by PacBio for validation by PCR. It would be helpful to confirm more alleles by PCR, particularly those with predicted mosaic sequences and those identified from single patients.

Response:

We chose to use only two alleles to obtain clear results for the downstream analyses. We could find specific primers for only three alleles that presented a suitable distribution among the patients to test their absence and presence. We think that these three specific PCRs clearly validate the procedure of identification of the alleles using PacBio followed by the dedicated bioinformatics pipeline.

2. Southern blots could help address some of the postulated mechanisms for generating diversity. While this is likely difficult to do with *P. jirovecii* due to the low amounts of organism DNA in clinical

samples, this could be done with other species, such as *P. carinii* or *P. murina*. Of note, duplication of subtelomeric regions has previously been identified in *P. carinii* and *P. murina*, supporting the proposed mechanism of rearrangement of the subtelomeres through single recombinations

Response:

Indeed, Southern blots are difficult and thus uncertain with *P. jirovecii*. Thus, it is extremely hazardous to use them to validate the structure of subtelomeres. Note that the rearrangement of subtelomeres through a single recombination that we hypothesize would lead to exchange, but not duplication as observed in *P. carinii* and *P. murina* (see Figure 8e).

3. It would be helpful to provide the gene IDs/GenBank numbers for the site-specific recombinase genes used to search of the Pneumocystis genome, and explain how the absence of this gene relates to the role of CRJE in recombination.

Response:

We have now added the IDs of the genes of site-specific recombinases used as baits at line 332 as follows:

“The UniProt identifiers of the baits used were P03870, P13769, P13770, P13783, P13784, and P13785.”

The sequence in mirror repeats within the CRJE could be a recognition sequence for site-specific recombinase because the latter recognize generally repeats (see Grindley et al 2006 Ann Rev Biochem 75:567). To clarify the issue, we have complemented the text at line 128 in the Supplementary data 6 as follows:

“Indeed, the CRJE includes a mirror repeat and such enzymes recognize generally repeats¹⁷.”

Thus, reference 17 has been added in the list for Supplementary data:

17. Grindley, N. D. F., Whiteson, K. L. & Rice, P. A. Mechanisms of site-specific recombination. Ann. Rev. Biochem. **75**, 567–695 (2006).

4. The use of the term “complete repertoire” in many places is confusing given that only a subset of the complete repertoires was identified from the majority of samples. It may be better to use a different term to accurately reflect the identified alleles.

Response:

Although they might be incomplete, as discussed above, we think that the term “complete” should be kept. Indeed, we have chosen it because these repertoires encompass both the non-expressed and the expressed alleles. We think that the text is clearer using “complete” than the possible alternative term “non-expressed + expressed”. We could use the term “non-expressed”, but this is misleading because it does not reflect that the expressed alleles are also included. However, if the Editor requires it, we could replace throughout the manuscript “complete” by “non-expressed”, and state at the first appearance of the latter that it corresponds in fact to both the non-expressed and expressed alleles.

5. The msg-I dataset generated in this study is potentially useful to the Pneumocystis research community for further investigation of msg diversity and evolution in different patient populations. Thus, it would be helpful to provide all msg-I alleles identified from each patient, especially for patients with a large number of msg-I alleles identified. If it's difficult to provide them as a supplemental file, they could be deposited into an online database.

Response:

The supplementary file providing the DNA sequences of all *msg-I* alleles identified in our study (*msg-I_alleles.fasta*) together with the supplementary file providing the alleles with their abundance in each patient (*msg-I_alleles_abundance_in_patients.xlsx*) permit to easily obtain the set of alleles present in each patient (using the free-share R software, for example). We think that depositing the necessary 48 datasets describing the alleles present each patient would constitute a useless duplicate (24 files for the complete repertoires, 24 for the expressed ones).

Reviewer #3 (Remarks to the Author):

This manuscript provides a large and impressive analysis of MSG gene diversity encoded by the important and genetically intractable fungus *Pneumocystis jirovecii*. The authors concentrate primarily on MSG family I, which has been proposed as being the most likely gene family involved in immune evasion by antigenic variation, given evidence that only one of an estimated 80 genes are adjacent to a promoter and, hence, likely to be transcribed at one time. Using PCR to recover expressed and silent copies of MSG 1 genes from 24 patients infected with *P. jirovecii*, the authors perform extensive analysis that reveals remarkable sequence variation, suggesting profound levels of change and reassortment. As such, the work provides very valuable insight into the MSG gene repertoire in this fungus and will act as an important reference dataset for understanding *Pneumocystis* immune evasion. I have a number of main concerns, and some smaller issues for the authors' consideration.

Main issues.

1. As stated above, this manuscript provides highly valuable information on MSG gene conservation and variation in static, geographically distributed samples. It cannot therefore directly inform on one aspect of immune evasion: MSG gene change over time during an infection, due to evasion of host adaptive immunity. It would be valuable for the authors to make this limitation clear in the discussion.

Response:

We agree and accordingly added the following sentence at line 813 in the section "Putting in perspective":

"...evolution over time remain to be determined, possibly by the analysis of sequential samples from the same patients. The analysis of such samples may also allow tackling a crucial aspect of immune evasion that our study has not investigated: the change of the Msg glycoproteins at the cell surface overtime."

2. The authors correctly state (eg line 317) that the PCR approach used may result in artefactual gene chimeras. They state that '[any] errors were specifically addressed and are not believed to affect the results presented below (see methods)'. However, the details of the tests run and provided in the methods is very limited and should be expanded upon. What two genes were tested; how similar were they; did the trimming resolve the issue with chimeras and, if so, how do we know it did? In addition to this, they state that only 5% of genes were pseudogenes (as defined by having stop codons); might the trimming have excluded truncated pseudogenes?

Response:

We chose to analyze a mix of two alleles to obtain clear results to facilitate downstream analysis. The two alleles used in the control experiment presented 62% identity, which is close to the average of 66% between all 1007 alleles observed in the study (their names and identity are now given in the Methods, see below). Our confirmation of the distribution of three alleles using specific PCRs followed by Sanger sequencing confirm the results obtained by PacBio sequencing followed by the dedicated bioinformatics pipeline. Furthermore, this is verified by the fact that the same alleles were observed in patients from various geographical locations (a mean of 84% of those composing the complete repertoires are present in at least one other patient). This is unlikely to result from the same chimeras formed because a different repertoire of alleles was present in each PCR.

The reads obtained by the control experiment upon mixing before PCR formed nine clusters. Two clusters corresponded to the two alleles present in the plasmids and totaled 92.5% of the reads (48.35% of PL1u100008324, 44.15% of PL1c172754267). Two other clusters represented 7.17% of the reads and gathered sequences with a single nucleotide polymorphism difference that were probably PacBio errors. The remaining five clusters included 205, 66, 28, 6, and 4 reads that were chimeras explained by a single recombination strictly in the first or last 500 bps, and that totaled 0.33% of all reads.

Accordingly, we have now expanded the methods at line 193 as follows:

“(plasmids PL1 and PL2 containing alleles PL1c172754267 and PL1u100008324 that show 62% identity, which is close to the average of 66% between all 1007 alleles observed in the study [see supplementary files msg-I alleles.fasta and msg-I alleles abundance in patients.xlsx]).”

Also, we have added at line 197:

“Most of the reads corresponded to the two alleles present in the plasmids (92.5%), 7.5% presented a single nucleotide polymorphism probably corresponding to a PacBio error, and 0.33% were chimeras explainable by a single recombination in the first or last 500 bps. These latter PCR artefacts result...”

The trimming resolved the issue of the chimeras that we did observe by our control experiment because there were strictly within the first and last 500 bps. Eventual other rare artefacts in the central part of the sequences would be difficult to detect. However, the fact that the same alleles are observed in patients from locations all over the world and our control with PCRs specific to three alleles assess that the alleles observed in our study are not chimeras (see here above our explanation).

The truncated pseudogenes have indeed been excluded by the selection of the PCR products of a size of ca. 3.100 bps carried out before PacBio sequencing and trimming of the sequences (described at line 164 of the Methods). Note that the trimming was done independently of the presence or not of an open reading frame, it was based only on the coordinates within the *msg-I* sequence. Consequently, it has not excluded entire pseudogenes.

Several aspects of the paper, and in particular the discussion, are very speculative and overlong, and a number of elements should be reduced and potentially discarded:

3. The authors suggest that MSG gene variation arises during meiosis; indeed, this is one of the main conclusions in the abstract. However, I cannot see any evidence to suggest MSG recombination is limited to, or even mainly arises during, meiosis, and so it appears to be merely a suggestion. I would suggest this focus should be reduced considerably.

Response:

We agree and accordingly have removed text and made modification as follows:

- At line 46 of the Abstract:
“.... mediation of homologous recombinations ~~during meiosis~~ by DNA triplexes.”
- At line 748 in the Discussion:

~~“These recombinations are likely to may occur mostly when the telomeres and subtelomeres are bundled as a “bouquet” by the attachment to the spindle body during the prophase of meiosis ^{15,47,48.}”~~

- At line 750 in the Discussion:

~~“This bouquet is involved in the matching and alignment of the homologous chromosomes by shaking them within the diploid cell. The latter phenomenon concerns most if not all eukaryotes, but has not been documented in the *Pneumocystis* genus so far.”~~

- At line 810 of the Discussion in the section “Putting in perspective”:

~~“This strategy relies on recombinations that take place probably mostly during the prophase of meiosis, within the bouquet of telomeres and subtelomeres. However, but their frequency of these recombinations and the speed of the subtelomeres evolution over....”~~

4. In the methods and the main text, the authors describe how they searched for recombinases, and refer the reader to supplementary data. No recombinases were found, and no detail is provided for what genes/proteins were looked for. As this is negative, it does not seem worth reporting.

Response:

Following the advice of reviewer 2, we have added the IDs of the site-specific enzymes used as baits in this search at line 332 of the Methods. We think it is worth to leave such negative results for eventual future reference, although as a supplementary data.

5. The description of potential triplex forming sequences (Fig. 7) is very interesting. However, there appears to be little/no evidence that these actually contribute to MSG recombination, or that they form the proposed secondary structures. As such, these data should come with a caution and be limited in their presentation. As a small point, the authors suggest the triplex may not be conserved in other *Pneumocystis* species, including *P. carinii*; I am not an expert, is this not the old name for *P. jirovecii*?

Response:

As we stated above in our response to reviewer 2, the presence of ca. 80 copies of the CRJE sequences in the subtelomeres is *per se* a strong evidence of the importance of this sequence, and of the mechanism it might imply. Consequently, we consider that its description and the discussion of this potentially crucial phenomenon deserve to be complete. This might foster fruitful research.

P. carinii was indeed used for all *Pneumocystis* organisms observed in different mammal hosts until the divergence at the genetic level between these different species was evidenced. Presently, *P. carinii* is used for one of the two *Pneumocystis* species infecting rat lungs.

6. ‘putting in perspectives’. This section could be removed, as it summarise what has been said before.

Response:

This section does not summarize what is said before, it puts our results into the epidemiological context of the disease caused by *P. jirovecii* (first and second paragraphs of this section). In the third paragraph of this section, we compare the strategy of antigenic

variation adopted by this fungus to those of other major human pathogens, which has not been mentioned before in the manuscript.

Thus, this section appears important to explain the context of our findings.

Small issues:

Line 97. This can be removed, as it is not clear what relevance the PhD thesis has to the paper.

Response:

This is asked by the University of Lausanne (but it could be removed if the Editors require).

Figure 2. I'm afraid I do not follow this graph, and cannot connect it to the previous data: what is denoted by 'number of strains (1-5)', and how does this relate to the patients and/or sample locations?

Response:

This Figure 2 shows the correlation between the number of strains infecting the patients and the number of alleles within the complete and expressed repertoire, with weighting by the number of PacBio reads of each observation. A significant correlation was observed only for the complete repertoires.

To clarify the issue, we have now added the two regression lines of these analyses on the Figure 2 and stated that in the legend. We also completed the legends of the axis as follows: "Number of strains per patient" and "Number of *P. jirovecii* msg-I alleles"

The relation with the provenance of the samples has not been investigated here because of the small numbers of patients from each location.

REVIEWER COMMENTS

Reviewer #1 (Remarks to the Author):

The authors have responded satisfactory to my comments and revised the manuscript appropriately.

Reviewer #2 (Remarks to the Author):

Many of the issues raised in the initial reviews have been addressed. However, there are some remaining concerns as noted below.

Major comments:

1. Given the absence of any experimental data, and the lack of consistent finding of the potential for triplex formation across other *Pneumocystis* species, we would strongly recommend removal of the reference to the triplex as a potential mechanism from the title at a minimum. Given that *Pneumocystis* genomes and CRJE sequences from additional host species are currently available, it would also be important to characterize the potential for triplex formation, as well as the potential RA target for proteases, in these additional species (in addition to *P. murina* and *P. carinii*). This analysis would help further clarify the potential specificity of triplex formation in *P. jirovecii* as suggested in the current manuscript.
2. The term "complete repertoire" as previously noted is inaccurate, as the authors have also acknowledged, thus retaining it is potentially misleading. Given this, we would strongly recommend revising the terminology, using for example "genomic" and "expressed" rather than "complete" and "expressed".
3. For the expressed repertoire, only a minority of *Msg* genes were identified (<15% of the complete repertoires in 13/24 patients, Table S5). This is in contrast to RNA-seq data in mice and rats, where in immunosuppressed animals all *Msg* genes are expressed. In addition to the potential biologic reasons for this summarized by the authors, and given that the genomic repertoires are incomplete, inefficient PCR or other technical issues may for a variety of reasons (primers not a match to all *Msg* sequences, inefficient amplification of certain *Msg* sequences, etc.) be operative. This should be noted in the potential reasons for the limited expressed repertoire (lines 663-678). Of note, especially of interest for future studies, the full-length sequence of the forward primer (GK135, originally used as a nested primer in the cited reference) exactly matches one or multiple regions in each of the 23 human chromosomes (GRCh38.p14 primary assembly), which potentially at least decreases its amplification efficiency for *msg* (due to insufficient amount of this primer to bind to the UCS region).
4. The authors suggest, in the Results (page 18, lines 428-430), that 84% of the alleles of each complete repertoire were shared with at least one other isolate. This statement is potentially misleading since 1. The identified genomic repertoires are incomplete, and 2. The *Msg* sequences were trimmed by ~1/3. This issue/concern should be identified/discussed in the manuscript, as it could result in an overestimate or underestimate of the level of sharing.

Minor concerns/suggestions:

1. Abstract: Given the limited geographic locations of patients whose samples were analyzed, replace "all over the world" with "between different countries".
2. Page 3, line 69: "8%" is not clearly supported by the references. More accurate estimates based on whole genome assemblies are available.
3. Page 4, lines 90-93: The concept that only one *msg-I* gene is expressed in a single cell at a given time remains hypothetical and has not been proven experimentally. Would revert to the original wording "... is believed to be expressed...".
4. Page 4, lines 98-99: Add "potential" in front of "mechanism".
5. Page 4, line 104: Since it's unknown if the model proposed is all inclusive, replace "complete" with

"propose", "hypothesize" or another similar word.

6. Page 7, line 171: Please clarify for how many samples this was necessary.

7. Page 28, line 641: Please clarify what is meant by "fully" and what data support that statement.

8. Page 30, lines 680-707: Given the uncertain role of DNA triplex in various organisms as highlighted in this paragraph and the absence of experimental evidence for *P. jirovecii* in this study, this section could be significantly condensed.

9. Table S4: If the sequences obtained in this study don't match any in GenBank, please deposit into GenBank and provide the accession number.

10. Table S7: Please clarify if the ITS1 region in this table is included in the ITS1-5.8S-ITS2 region in Table S5 and if the results are consistent between these two loci. Provide GenBank # if the sequences don't match any in GenBank.

Reviewer #3 (Remarks to the Author):

The authors have provided substantial responses to my suggestions, improving an interesting article that provides new information in this field.

RESPONSES TO THE REVIEWERS' COMMENTS, SECOND REVISION

(NCOMMS-23-15687A, Meier et al.)

Reviewer #2 (Remarks to the Author):

Major comments:

1. Given the absence of any experimental data, and the lack of consistent finding of the potential for triplex formation across other *Pneumocystis* species, we would strongly recommend removal of the reference to the triplex as a potential mechanism from the title at a minimum.

Response:

Accordingly we have now removed “ ..., potentially mediated by DNA triplexes ” from the manuscript’s title.

1 (continued). Given that *Pneumocystis* genomes and CRJE sequences from additional host species are currently available, it would also be important to characterize the potential for triplex formation, as well as the potential RA target for proteases, in these additional species (in addition to *P. murina* and *P. carinii*). This analysis would help further clarify the potential specificity of triplex formation in *P. jirovecii* as suggested in the current manuscript.

Response:

We agree and have now modified the text as follows from line 607 at the end of the results section, and added the new Supplementary data 7 as well as the new Figure S13 (**NB:** in the revised manuscript, new text is underlined, deleted text in struck through) :

~~This may have evolved specifically within the human host because the CRJE sequences of *P. carinii* and *P. murina* do not encode such motif.~~

As far as the other species of the genus are concerned, *Pneumocystis macacae* and *Pneumocystis oryctolagi* harbours a CRJE sequence similar to that of *P. jirovecii*, including the presence of two R residues in addition to that present in the recognition site of the Kexin (Supplementary data 7, Figure S13). Thus, these CRJEs might also form a DNA triplex and be recognized by a further protease. On the other hand, nine other *Pneumocystis* species harbour a CRJE more distant from the canonical mirror repeat forming *H-DNA triplexes. However, they might anyway form such structure because the formation of *H-DNA triplex is more versatile and less requiring at the level of sequence than canonical H-DNA²⁸.

~~As far as the other species of the genus are concerned, *P. carinii* harbours a CRJE sequence with a less symmetrical mirror repeat than that of *P. jirovecii*⁴⁷, but that could possibly form a DNA triplex (not shown). On the other hand, the one present in *P. murina* presents a much less conserved mirror repeat and symmetry^{47,36}, so that formation of triplex appears unlikely.~~

Supplementary data 7 :

7. **Comparison of the CRJE sequences of different *Pneumocystis* species**

We investigated if the CRJE sequence of other species than *P. jirovecii* can also potentially form *H-DNA triplex and encode several R residues. Eleven CRJEs are shown in Figure S13a and the alignments of their DNA sequences and encoded peptides are shown with their corresponding phylogenetic trees in Figure S13b. The trees are consistent with those previously reported on the basis on the entire UCS including the CRJE²⁰ or on 106 single-copy genes²¹ (in the latter one, *P. wakefieldiae* is close to *P. carinii* and *P. murina* possibly because of the absence of *P. sp. "exulans"* and *P. sp. "tanezumi"*). These trees reveal three groups of CRJEs and one outlier.

Group 1 includes CRJEs of two *Pneumocystis* species specific to primates and one to rabbit (*P. jirovecii*, *P. macacae*, *P. oryctolagi*). Although that of *P. oryctolagi* is less symmetrical, the three CRJE sequences might potentially form *H-DNA because they closely resemble the canonical mirror repeat reported to do so (see Results, section "Structure of the CRJE sequence present at the beginning of each *msg-I* gene"). The peptide encoded by the CRJE of *P. jirovecii* presents a repetition of the motif ARAV that is not observed in those of *P. macacae* and *P. oryctolagi*. However, the two latter present also two R residues in addition to that present in the Kexin recognition site KR that might be recognized by a further protease. This might further ensure the proper removal of the constant part of the *Msg-I* proteins that is believed to be carried out by the Kexin at the site KR (see Results). Interestingly, these two R residues are encoded by a codon including bases that are imperfect in the mirror repeat, as in *P. jirovecii*.

Group 2 includes CRJEs of three *Pneumocystis* species specific to different rat species and one to mouse (*P. carinii*, *P. murina*, *P. sp. "fulvescens"*, *P. sp. "muelleri"*). These CRJEs are less likely to form *H-DNA triplex because they present less symmetry and no clear mirror repeat as compared to the canonical sequence. Nevertheless, the strand shown is enriched in purines suggesting that formation of triplex might occur. Indeed, the formation of *H-DNA triplex is more versatile and less requiring at the level of sequence than canonical H-DNA²². Out of the four CRJEs of this group, only that of *P. carinii* presents a supplementary R residue in the encoded peptide in addition to the KR site.

Group 3 includes CRJEs of three *Pneumocystis* species specific to different rat species (*P. wakefieldiae*, *P. sp. "tanezumi"*, *P. sp. "exulans"*). These CRJEs are more distant from the canonical mirror repeat than those of group 2, suggesting that formation of *H-DNA triplex by them is even less likely. Nevertheless, they present each a stretch enriched in purines that may form a triplex according to Mirkin²². They do not present supplementary R residues in their encoded peptide.

The outlier CRJE of *P. canis* encodes one supplementary R residue.

Legend of Figure S13 :

Figure S13. Features of the CRJE of 11 *Pneumocystis* species that is present at the end of the UCS and at the beginning of each *msg-I* gene. The specific mammalian host is given under the *Pneumocystis* species name. The CRJE sequences are those present in Figure 4 of Ma et al²⁰. Their NCBI GenBank accession numbers are: *P. jirovecii* (T551_00002), *Pneumocystis sp. "macacae"* (MN509821), *Pneumocystis oryctolagi* (MN507527), *P. carinii* (T552_04149), *P. murina* (PNEG_04309), *Pneumocystis sp. "fulvescens"* (MN509819), *Pneumocystis sp. "muelleri"* (MN509817), *Pneumocystis wakefieldiae* (AF164562), *Pneumocystis sp. "tanezumi"* (MN509820), *Pneumocystis sp. "exulans"* (MN509818), and *Pneumocystis canis* (MN509823). In absence of the sequences of several *msg-I* genes, the 5' extremity of each CRJE is approximate for all species except *P.*

jirovecii, *P. carinii*, *P. murina* and *P. oryctolagi*. The sequence of the CRJE of *P. murina* shown is only the 44 out of 132 bps that are left of the Kexin site²⁴.

- (a) The strand of the CRJE shown is enriched in purines and encodes the peptide given underneath that is part of the Msg-I protein. For group 1, the mirror repeat is symbolized by the convergent arrows and the imperfect positions are underlined. The recognition site of the Kexin KR is underlined as well as its encoding codons. Cytosines at imperfect positions of the mirror repeat lead to all R residues present in the peptides in addition to that present in the site KR. The percentage of purines within the enriched stretch(es) identified arbitrarily by arrows or lines above the nucleotide sequence are given on the right.
- (b) The alignments of the DNA sequences and encoded peptides of the 11 CRJEs were obtained using Clone Manager 9 Professional Edition software version 9.51 (Sci Ed Software LLC) and the “similarity format” with areas of high matches colored in blue. The linear scoring matrix was used for the corresponding trees shown.

2. The term “complete repertoire” as previously noted is inaccurate, as the authors have also acknowledged, thus retaining it is potentially misleading. Given this, we would strongly recommend revising the terminology, using for example “genomic” and “expressed” rather than “complete” and “expressed”.

Response:

We have now replaced “complete” by “genomic” throughout the manuscript.

3. For the expressed repertoire, only a minority of Msg genes were identified (<15% of the complete repertoires in 13/24 patients, Table S5). This is in contrast to RNA-seq data in mice and rats, where in immunosuppressed animals all Msg genes are expressed. In addition to the potential biologic reasons for this summarized by the authors, and given that the genomic repertoires are incomplete, inefficient PCR or other technical issues may for a variety of reasons (primers not a match to all Msg sequences, inefficient amplification of certain Msg sequences, etc.) be operative. This should be noted in the potential reasons for the limited expressed repertoire (lines 663-678).

Response:

We have now modified the text at lines 696-701 of the discussion as follows :

- (iv) The technical limitations of the methodology may have played a role in the generally low proportion of the genomic repertoires present in the expressed repertoires (<15% in 13 of the 24 patients, Table S5). Indeed, the number of alleles in the expressed repertoires may have been underestimated. This might explain that our observations contrast with the 100% of genomic *msg-I* genes being expressed in *P. carinii* and *P. murina* infections⁴. Varying selective pressures by the different hosts may also have been involved.

3 (continued). Of note, especially of interest for future studies, the full-length sequence of the forward primer (GK135, originally used as a nested primer in the cited reference) exactly matches one or multiple regions in each of the 23 human chromosomes (GRCh38.p14 primary assembly), which potentially at least decreases its amplification efficiency for msg (due to insufficient amount of this primer to bind to the UCS region).

Response:

Thank you. This caveat is now mentioned at line 40 of the Supplementary data 1 as follows :

“Moreover, in the case of the expressed repertoires, the efficiency of amplification has probably been decreased by the fact that the forward primer GSK135 within the UCS matches multiple regions of the human genome.”

4. The authors suggest, in the Results (page 18, lines 428-430), that 84% of the alleles of each complete repertoire were shared with at least one other isolate. This statement is potentially misleading since 1. The identified genomic repertoires are incomplete, and 2. The Msg sequences were trimmed by ~1/3. This issue/concern should be identified/discussed in the manuscript, as it could result in an overestimate or underestimate of the level of sharing.

Response:

This caveat is now mentioned at line 431 of the Results as follows :

“Of note, this level of sharing might be imprecise because of the trimming of the sequences and the possible underestimation of the number of alleles present in the genomic repertoires (see previous section). However, this would not impact the conclusions drawn as they are not based on the absolute values.”

Minor concerns/suggestions:

1. Abstract: Given the limited geographic locations of patients whose samples were analyzed, replace “all over the world” with “between different countries”.

Response:

This is now replaced as suggested at line 52.

2. Page 3, line 69: “8%” is not clearly supported by the references. More accurate estimates based on whole genome assemblies are available.

Response:

This is now modified as follows at line 70 of the Introduction :

“it represents up to 6% ~~ca. 8%~~ of its highly compacted genome^{5,6,7}”.

Accordingly, the following reference has been added as no. 5 and the references have been renumbered accordingly :

5. Cissé, O.H. et al. Genomic insights into the host specific adaptation of the *Pneumocystis* genus. Com. Biol. 4, 305 (2021).

3. Page 4, lines 90-93: The concept that only one msg-I gene is expressed in a single cell at a given time remains hypothetical and has not been proven experimentally. Would revert to the original wording "... is believed to be expressed...".

Response:

This is now reverted as suggested at line 92.

4. Page 4, lines 98-99: Add "potential" in front of "mechanism".

Response:

This is now added as suggested at line 99.

5. Page 4, line 104: Since it's unknown if the model proposed is all inclusive, replace "complete" with "propose", "hypothesize" or another similar word.

Response:

This is now replaced as suggested at line 105.

6. Page 7, line 171: Please clarify for how many samples this was necessary.

Response:

"Some" is now replaced by "38 out of the 48" at line 173.

7. Page 28, line 641: Please clarify what is meant by "fully" and what data support that statement.

Response:

To clarify the issue, we have modified the sentence from line 657 as follows:

" These recombinations might occur preferentially in the 33 bps CRJE sequence present at the beginning of each CDS and the 31 bps sequence located after the stop codon because these sequences are entirely fully conserved within the up- and downstream regions all *msg-I* gene sequences reported to date. "

8. Page 30, lines 680-707: Given the uncertain role of DNA triplex in various organisms as highlighted in this paragraph and the absence of experimental evidence for *P. jirovecii* in this study, this section could be significantly condensed.

Response:

Accordingly, this paragraph has been reduced from 339 to 184 words from line 705 as follows :

~~“The conservation of the CRJE sequence *in toto* and in multiple copies in the subtelomeres, precisely at the location where recombinations leading to the exchange the expressed allele are postulated, very strongly suggest that it plays a crucial role in the antigenic variation system of *P. jirovecii*. The DNA triplexes that these sequences can potentially form because of their distinct motif including an imperfect mirror could be involved. However, despite that they mirror repeats constitute a hallmark representing up to 1% of eukaryotes’ genomes³⁶⁻³⁸, their function of the imperfect mirror repeats has been difficult to assess and remains putative so far. This situation results from the fact that DNA triplexes are notoriously difficult to tract, mostly because of their putative transient state and that the conditions required for their formation are not reproduced easily *in vitro*^{29,31,39}. Nevertheless, a large body of circumstantial evidence⁴⁰, mainly their location within the genome, suggests that they are involved in a number of genetic processes⁴⁰ : transcription, replication, chromosome folding, structure of chromosome ends, mutational process, instability, rearrangements, translocations, and homologous recombination. In our context, the latter three processes are highly relevant, and Homologous recombination is in fact the function process that was most recurrently mentioned because mirror repeats have often been reported close to recombination sites^{29,35,40-43}. Moreover, mediation of homologous recombination is the only potential function that has been supported by an experimental evidence: the presence of a polypurine-polypyrimidine stretch within a plasmid of *Escherichia coli* enhanced homologous recombinations within repetitive sequences present nearby⁴⁴. Furthermore, nucleic acid triplexes (RNA:DNA hybrids, R-loops) would be involved in (i) the switching of the expressed allele participating to the antigenic variation system of *Trypanosoma brucei*⁴⁵, and (ii) the switch recombinations in mammalian immunoglobulins⁴⁶. Thus, we hypothesize that the DNA triplexes potentially formed by the CRJE sequences mediate, perhaps activate, the recombinations involved in the translocations of the *msg-I* genes, and possibly also the other recombinations involved in the system of antigenic variation of *P. jirovecii*. “~~

Thus, this paragraph now reads :

“The conservation of the CRJE sequence *in toto* and in multiple copies in the subtelomeres strongly suggest that it plays a crucial role in the antigenic variation system of *P. jirovecii*. The DNA triplexes that these sequences can potentially form could be involved. However, despite that mirror repeats constitute a hallmark representing up to 1% of eukaryotes’ genomes³⁶⁻³⁸, their function remains putative so far. This situation results from the fact that DNA triplexes are notoriously difficult to tract^{29,31,39}. Nevertheless, a large body of circumstantial evidence suggests that they are involved in a number of genetic processes⁴⁰. Homologous recombination is the process that was most recurrently mentioned because mirror repeats have often been reported close to recombination sites^{29,35,40-43}. Furthermore, nucleic acid triplexes (RNA:DNA hybrids, R-loops) would be involved in (i) the switching of the expressed allele participating to the antigenic variation system of *Trypanosoma brucei*⁴⁵, and (ii) the switch recombinations in mammalian immunoglobulins⁴⁶. Thus, we hypothesize that the DNA triplexes potentially formed by the CRJE sequences mediate, perhaps activate, the recombinations involved in the system of antigenic variation of *P. jirovecii*. “

9. Table S4: If the sequences obtained in this study don't match any in GenBank, please deposit into GenBank and provide the accession number.

Response:

The sequences have been now deposited in Genbank and their accessions numbers provided in the Data availability statement at line 836 as follows:

Data availability statement

PacBio CCS raw reads (*msg-I*: accession nos. SRR24284242 to SRR24284301, ITS1-5.8S-ITS2: SSR25739987 to SRR25740015) have been deposited in the NCBI Sequence Read Archive linked to BioProject accession no. PRJNA936793 and BioSample accession no. SAMN33368625. The sequences obtained in this study have been deposited in GenBank (1007 new *msg-I* alleles: accession nos. OR489167 to OR490173; 15 new ITS1-5.8S-ITS2 alleles: OR475686 to OR475700). ~~The identified *msg-I* alleles and a table including their relative abundance of each *msg-I* allele identified in each patient are~~ is provided as a supplementary files (msg-I_alleles.fasta, msg-I_alleles_abundance_in_patients.xlsx).

The accession numbers of the 15 new ITS1-5.8S-ITS2 alleles are also given in Table S4 and the following footnote has been added for each of them:

^b This study.

The supplementary files msg-I_alleles.fasta and ITS1-5.8S-ITS2_alleles.fasta have been deleted because the sequences are now available from GenBank.

10. Table S7: Please clarify if the ITS1 region in this table is included in the ITS1-5.8S-ITS2 region in Table S5 and if the results are consistent between these two loci. Provide GenBank # if the sequences don't match any in GenBank.

Response:

Accordingly, the following footnote has been added to Table S7 :

^b The ITS1 sequences observed were identical to that observed by PacBio sequencing (Table S5), except sample LA9 that harboured in addition allele B2.

REVIEWERS' COMMENTS

Reviewer #2 (Remarks to the Author):

The authors have addressed all our remaining issues.